# Myeloid cell deficiency of p38γ/p38δ protects against candidiasis and regulates antifungal immunity

Dayanira Alsina-Beauchamp[1], Alejandra Escós[1] (iD), Pilar Fajardo[1], Diego González-Romero[1], Ester Díaz-Mora[1], Ana Risco[1], Miguel A Martín-Serrano[1], Carlos del Fresno[2], Jorge Dominguez-Andrés[1], Noelia Aparicio[1], Rafal Zur[1], Natalia Shpiro[3], Gordon D Brown[4], Carlos Ardavín[1], Mihai G Netea[5], Susana Alemany[6], Juan J Sanz-Ezquerro[7] & Ana Cuenda[1,*] (iD)

## Abstract

*Candida albicans* is a frequent aetiologic agent of sepsis associated with high mortality in immunocompromised patients. Developing new antifungal therapies is a medical need due to the low efficiency and resistance to current antifungal drugs. Here, we show that p38γ and p38δ regulate the innate immune response to *C. albicans*. We describe a new TAK1-TPL2-MKK1-ERK1/2 pathway in macrophages, which is activated by Dectin-1 engagement and positively regulated by p38γ/p38δ. In mice, p38γ/p38δ deficiency protects against *C. albicans* infection by increasing ROS and iNOS production and thus the antifungal capacity of neutrophils and macrophages, and by decreasing the hyper-inflammation that leads to severe host damage. Leucocyte recruitment to infected kidneys and production of inflammatory mediators are decreased in p38γ/δ-null mice, reducing septic shock. p38γ/p38δ in myeloid cells are critical for this effect. Moreover, pharmacological inhibition of p38γ/p38δ in mice reduces fungal burden, revealing that these p38MAPKs may be therapeutic targets for treating *C. albicans* infection in humans.

**Keywords** *Candida albicans*; infection; kinase inhibitor; p38MAPK; signalling
**Subject Categories** Immunology; Microbiology, Virology & Host Pathogen Interaction; Pharmacology & Drug Discovery

## Introduction

*Candida* (*C.*) *albicans* is a harmless component of the human microbiota; however, under conditions in which tissue homeostasis is altered and host defence is compromised, *C. albicans* becomes a pathogen and can invade the mucosa reaching the bloodstream and causing systemic infection (Brown *et al*, 2012; Netea *et al*, 2015; Kim, 2016). Invasive fungal infection by *Candida* species (spp) is a serious health concern, particularly for immunocompromised patients (Brown *et al*, 2012; Kim, 2016). Among *Candida* spp, the sepsis caused by *C. albicans* is one of the most frequent in hospital intensive care units in patients with AIDS or auto-immune diseases and in those undergoing anti-cancer chemotherapy or organ transplantation (Wisplinghoff *et al*, 2006; Brown *et al*, 2012; Kullberg & Arendrup, 2015, 2016). In the last years, the cases of invasive candidiasis have increased and the mortality rate associated with it is higher than 40%, even in patients receiving antifungal therapy (Brown *et al*, 2012; Kullberg & Arendrup, 2015, 2016). The development of novel antifungal drugs is insufficient, and only a small number are currently used in clinical applications. Moreover, antifungal drug resistance is worryingly growing (Kim, 2016). Therefore, understanding how the immune response towards *C. albicans* is mounted and knowing the mechanism of immune resistance to fungal spread is crucial to develop novel therapeutic strategies to combat candidiasis.

Macrophages and neutrophils are at the first line of defence against *C. albicans* and are important for the activation and regulation of the innate immune response. These cells express pattern recognition receptors (PRR) that recognize molecules on the surface of the invading pathogens, called pathogen-associated molecular patterns (PAMP; Lee & Kim, 2007). Activation of PRRs on innate immune cells leads to the secretion of cytokines and other mediators that promote the elimination of infectious agents and induction of tissue repair (Lee & Kim, 2007). The best-characterized PRRs are the Toll-like receptors (TLR; Takeda *et al*, 2003) and the C-type lectin receptors (CLR; Netea *et al*, 2008). Both are involved in *C. albicans* recognition by binding to different molecules on the fungal surface.

1 Department of Immunology and Oncology, Centro Nacional de Biotecnología/CSIC, Madrid, Spain
2 Immunobiology of Inflammation Laboratory, Centro Nacional de Investigaciones Cardiovasculares Carlos III, Madrid, Spain
3 Medical Research Council Protein Phosphorylation Unit, Sir James Black Building, School of Life Sciences, University of Dundee, Dundee, UK
4 Aberdeen Fungal Group, Institute of Medical Sciences, Medical Research Council Centre for Medical Mycology at the University of Aberdeen, Aberdeen, UK
5 Department of Internal Medicine and Radboud Center for Infectious Diseases, Radboud University Nijmegen Medical Centre, Nijmegen, The Netherlands
6 Instituto de Investigaciones Biomédicas Alberto Sols, CSIC-UAM, Madrid, Spain
7 Department of Cellular and Molecular Biology, CNB/CSIC, Madrid, Spain
*Corresponding author. Tel: +34-915855451; E-mail: acuenda@cnb.csic.es

In macrophages, TLR2 and TLR4 recognize phospholipomannans, whereas the CLR Dectin-1 recognizes β-glucans (Netea *et al*, 2015). Upon fungus recognition, the stimulation of these receptors triggers the activation of NF-κB and MAPK pathways, which are crucial to generate the immune responses (Lee & Kim, 2007; Geijtenbeek & Gringhuis, 2009; Takeuchi & Akira, 2010). The three major MAPK pathways activated by PRRs are those leading to the activation of c-Jun N-terminal Kinase (JNK), p38MAPK and extracellular signal-regulated kinase 1/2 (ERK1/2; Lee & Kim, 2007; Gaestel *et al*, 2009).

There are four p38MAPK isoforms, p38α, p38β, p38γ, and p38δ, encoded by different genes, which are activated in response to a range of cell stresses and in response to inflammatory cytokines (Cuenda & Sanz-Ezquerro, 2017). Among them, p38α is crucial for the synthesis of pro-inflammatory molecules and for the regulation of the immune response (Gaestel *et al*, 2009). Also, p38γ and p38δ (p38γ/p38δ) have recently been shown to play important roles in regulating cytokine production, T-cell activation, insulin resistance, and in tumorigenesis associated with inflammation (Risco *et al*, 2012; Criado *et al*, 2014; Escós *et al*, 2016; Cuenda & Sanz-Ezquerro, 2017). Although several studies have demonstrated that p38γ/p38δ are involved in inflammatory processes, the functional roles of these kinases in the innate immune responses have not been fully characterized. In particular, the role of p38γ/p38δ in *C. albicans* infection is completely unknown. We have therefore investigated the role of p38γ/p38δ on the *C. albicans*-mediated activation of macrophages and in the early innate response against fungal infection, using a mouse model of systemic candidiasis in which the kidney is the main target organ (Lionakis *et al*, 2013). Here we report that p38γ/p38δ are important in modulation of host antifungal immune response, since their deletion, particularly in myeloid cells, protects against *C. albicans* infection and increases fungal elimination by neutrophils and macrophages. We also identified a novel signalling pathway downstream of Dectin-1 in which TAK1-IKKβ-TPL2 are essential for ERK1/2 activation and cytokine production in mouse macrophages and human monocytes. TPL2 is the upstream kinase that mediates MKK1-ERK1/2 activation after TLR stimulation, whose protein level is regulated by p38γ/p38δ (Risco *et al*, 2012). Finally, we show that *in vivo* pharmacological p38γ/p38δ inhibition, using kinase inhibitors, reduces the symptoms of *C. albicans* systemic infection and increases the clearance of the fungus in the kidney. This study could be the basis for designing novel therapeutic strategies in invasive candidiasis using p38γ/p38δ as targets.

# Results

### p38γ/p38δ regulate *Candida albicans*-induced cytokine production

To investigate the role of p38γ/p38δ in *C. albicans* infection, we assessed inflammatory cytokine and chemokine mRNA levels in response to heat-killed *C. albicans* (HK-Ca) in WT and p38γ/δ$^{-/-}$ bone marrow-derived macrophages (BMDM) *in vitro*. Macrophages are one of the main cells that come in contact with the fungus early after infection in candidiasis (Lionakis *et al*, 2013). HK-Ca stimulation in p38γ/δ$^{-/-}$ BMDM had no effect in the mRNA

expression of *TNFα* and *IL-6*, whereas *IL-1β*, *IL-10*, *KC*, *MIP-2* and *CCL2* mRNA expression was markedly reduced as compared to that in WT BMDM (Fig 1A), indicating the need of these p38 kinases for cytokine and chemokine production in the response to *C. albicans*.

To assess whether p38γ/p38δ loss altered the signalling response involved in cytokine production, we analysed changes in NF-κB and MAPK (p38α, ERK1/2, JNK) pathways activation in response to *C. albicans* in p38γ/δ$^{-/-}$ and WT BMDM. We found that p38γ/p38δ deletion did not have a significant effect on the activation of p38α and the NF-κB pathways, although the activation of these pathways was more sustained in WT than in p38γ/δ$^{-/-}$ BMDM (Fig 1B). In contrast, ERK1/2 phosphorylation was substantially reduced in p38γ/δ-null BMDM as compared to WT BMDM (Fig 1B). We did not detect significant HK-Ca-induced phosphorylation of JNK1/2 either in WT or p38γ/δ$^{-/-}$ BMDM (Fig 1B).

To determine the mechanism by which ERK1/2 pathway activation is impaired in p38γ/δ$^{-/-}$ BMDM after *C. albicans* infection, we examined ERK1/2 phosphorylation in response to specific ligands for each *C. albicans*-activated receptor in macrophages. We have previously reported that p38γ/p38δ positively regulate TLR4-induced ERK1/2 activation and cytokine production by controlling the steady-state levels of the MKK kinase TPL2 in macrophages (Risco *et al*, 2012), which is not expressed in p38γ/δ$^{-/-}$ cells (Fig 1B). p38γ/δ deletion impaired ERK1/2 pathway activation in response to both the TLR2/6 ligand tripalmitoylated lipopeptide (Pam3Cys) and the TLR4 ligand LPS (Fig 1C). Accordingly, the activation of ERK1/2 pathway in response to unmethylated CpG oligonucleotide (ODN, TLR9-ligand), Imiquimod (TLR7-ligand) and poly I-C (TLR3-ligand), which is mediated by TPL2, was also impaired in p38γ/δ$^{-/-}$ macrophages (Appendix Fig S1A and B). In contrast, ERK1/2 activation in response to PMA, which is dependent on Raf-1, another MKK1/2 kinase (Wellbrock *et al*, 2004), and TPL2-independent (Beinke & Ley, 2004) was unaffected (Appendix Fig S1B). p105 NF-κB1, JNK1/2 and p38α were still phosphorylated in response to TLR ligands in p38γ/δ-null macrophages (Appendix Fig S1C). All these data demonstrate that p38γ/p38δ regulate ERK1/2 activation triggered by TLRs. Nonetheless, we also examined the contribution of Dectin-1 to cytokine production and ERK1/2 pathway activation in BMDM by stimulating with Curdlan, a purified β-glucan that mimics fungal stimulation in innate immune cells (Gantner *et al*, 2003). The stimulation of p38γ/δ-null macrophages with Curdlan resulted in decreased IL-1β production (Fig 1D) and decreased MKK1-ERK1/2 activation as compared to WT BMDM (Fig 1E). p38α, JNK1/2 and p105 NF-κB1 were phosphorylated in response to Curdlan in WT and p38γ/δ$^{-/-}$ macrophages (Fig 1E).

We confirmed the specificity of Curdlan as a Dectin-1 ligand by examining ERK1/2 activation in MyD88$^{-/-}$, Dectin1$^{-/-}$ BMDM and also using the Syk inhibitor PRT062607 (Spurgeon *et al*, 2013) in WT BMDM. TLRs signal via the adaptor molecule MyD88, whereas Dectin-1 signalling is mediated by the recruitment of the tyrosine kinase Syk (Underhill, 2007; Reid *et al*, 2009; Takeuchi & Akira, 2010). Curdlan, HK-Ca and LPS (used as control) induced ERK1/2 phosphorylation in WT macrophages. The activation of ERK1/2 in MyD88$^{-/-}$ BMDM was blocked in response to LPS, whereas in response to Curdlan was not affected and in response to HK-Ca was partially impaired, compared to WT BMDM (Appendix Fig S1D). Furthermore, in Dectin1$^{-/-}$ BMDM,

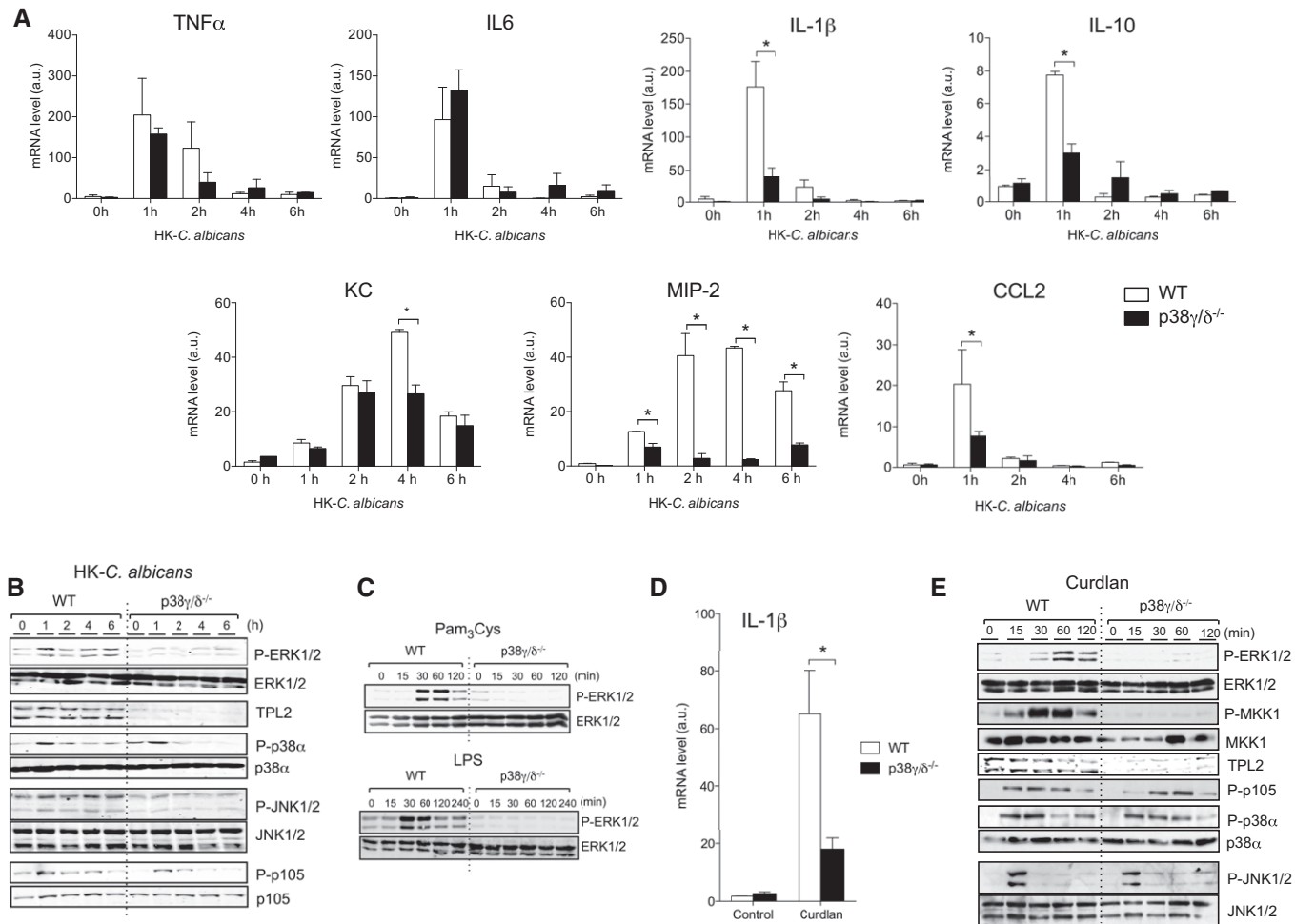

**Figure 1.  Cytokine production and ERK1/2 activation is impaired in p38γ/p38δ-null BMDM in response to TLRs and Dectin-1 ligands.**

A   BMDM from WT or p38γ/δ$^{-/-}$ mice were exposed to HK-Ca (1 × 10$^6$ CFU/ml) for the indicated times. Relative mRNA expression was determined by qPCR for TNFα, IL-6, IL-1β, IL-10, KC, MIP-2 and CCL2. Results were normalized to β-actin mRNA expression, and x-fold induction was calculated relative to WT expression at 0 h. Data show mean ± SEM from one representative experiment of two in triplicate, with similar results. Only significant results are indicated, *$P ≤ 0.05$ relative to WT BMDM exposed to HK-Ca, at each time point. Parametric, unpaired *t*-test.

B   BMDM from WT or p38γ/δ$^{-/-}$ mice were stimulated as in (A). Cell lysates (50 μg) were immunoblotted with antibodies to active phosphorylated ERK1/2 (P-ERK1/2), p38α (P-p38α) or JNK1/2 (P-JNK1/2), or to phosphorylated p105 NF-κB1 (P-p105). Total protein levels for the above proteins and for TPL2 were also measured as loading controls. Representative immunoblots from three independent experiments are shown.

C   BMDM were stimulated 200 ng/ml Pam3Cys or 100 ng/ml LPS. Cell lysates were immunoblotted with the indicated antibodies. Representative immunoblots from three independent experiments are shown.

D   BMDM were exposed for 1 h to 10 μg/ml Curdlan. Relative mRNA expression was determined by qPCR for IL-1β. Results were normalized and fold induction calculated as in (A). Data show mean ± SEM from one representative experiment of two in triplicate, with similar results. Only significant results are indicated, *$P ≤ 0.05$ relative to WT BMDM exposed to Curdlan. Parametric, unpaired *t*-test.

E   BMDM were stimulated with 10 μg/ml Curdlan and cell lysated immunoblotted as shown. Representative immunoblots from three independent experiments are shown.

Source data are available online for this figure.

ERK1/2 activation by Curdlan was blocked, by HK-Ca was partially inhibited and by LPS was not affected (Appendix Fig S1E). In WT macrophages pre-treated with the compound PRT062607, the phosphorylation of ERK1/2 was significantly inhibited in response to Curdlan and was not affected in response to LPS (Appendix Fig S1F). These results confirm that, in macrophages, the activation of ERK1/2 induced by Curdlan was mediated by Dectin-1-Syk signalling and not by TLRs, whereas ERK1/2 activation in response to HK-Ca was also partially induced via TLR-MyD88 signalling.

## TAK1-IKKβ-TPL2 activation is essential for Dectin-1 signalling in macrophages

Since TPL2 was not expressed in p38γ/δ$^{-/-}$ cells (Fig 1B and E), our results suggest that the kinase TPL2 regulates ERK1/2 activation triggered by Dectin-1. To investigate this in more depth, we used TPL2$^{-/-}$ BMDM. In these cells, ERK1/2 activation was abolished in response to LPS, Curdlan, HK-Ca and Zymosan, which activates Dectin-1 and TLR2 in macrophages (Brown *et al*, 2003;

Fig 2A), confirming that TPL2 mediates ERK1/2 activation not only downstream of TLR but also downstream of the receptor Dectin-1. Consistently, we found that pharmacological blockade of TPL2 by the compound C34 (Green *et al*, 2007; Wu *et al*, 2009; Appendix Fig S2A) and of MKK1 by the inhibitor PD184352 (Bain *et al*, 2007) inhibited Curdlan and HK-Ca-induced ERK1/2 activation in WT BMDM (Fig 2B and Appendix Fig S2B). C34 also reduced *IL-1β* mRNA production in Curdlan- and HK-Ca-activated WT BMDM, but not in p38γ/δ$^{-/-}$ BMDM (Fig 2C and Appendix Fig S2C), supporting the important role of p38γ/p38δ-TPL2 in Dectin-1 signalling. This is the first time that a role for TPL2 in ERK1/2 activation triggered by Dectin-1 signalling is described.

TPL2 activation is regulated by IKKβ in TLR-stimulated macrophages (Gantke *et al*, 2011; Roget *et al*, 2012; Ben-Addi *et al*, 2014). Moreover, the kinase TAK1 is required for the activation of IKKβ in the canonical IKK complex leading to TPL2-MKK1-ERK1/2 activation (Cohen, 2014). Therefore, to investigate whether the TAK1-IKK signalling pathway was involved in the activation of ERK1/2 induced by Dectin-1, we treated macrophages with the highly selective IKKβ inhibitor BI605906 (Clark *et al*, 2011) or with the potent TAK1 inhibitor, NG25 (Dzamko *et al*, 2012). BMDM pre-treatment with BI605906 or NG25 blocked Curdlan from inducing ERK1/2 activation (Fig 2D and E). Accordingly, BI605906 and NG25 also impaired ERK1/2 phosphorylation in response to HK-Ca (Fig 2E and F). NG25 also inhibited Curdlan- and HK-Ca-induced phosphorylation of p105 NF-κB1 and in response to LPS the ERK1/2 and p105 NF-κB1 phosphorylation (Fig 2E). All our data together indicate that p38γ/p38δ positively modulate TPL2 levels and that TAK1-IKKβ-TPL2 regulate MKK1-ERK1/2 activation in both Dectin-1 and TLR pathways in macrophages (Fig 2G).

Additionally, we studied the implication of TPL2 in *C. albicans* response in human peripheral blood mononuclear cells (PBMCs)-derived monocytes. In these cells, Dectin-1 is an important receptor for the immune sensing of *C. albicans*, since incubation with laminarin, a specific Dectin-1 inhibitor, largely decreases cytokine production in response to the fungus (Toth *et al*, 2013). Monocytes were pre-treated with the TPL2 inhibitor C34 or the p38α/β inhibitor SB203580, as control, and then stimulated with HK-Ca. As expected, HK-Ca-induced ERK1/2 phosphorylation was reduced to the basal levels by pre-incubation with C34 (Appendix Fig S2D). HK-Ca-induced expression of TNFα, IL-6 and IL-10 was impaired by C34, whereas SB203580 only partially blocked IL-10 production (Appendix Fig S2E), supporting the conclusion that *in vitro*, *C. albicans*-induced cytokine production is dependent on TPL2 activity also in human monocytes.

## p38γ/p38δ deletion protects from *Candida albicans* infection

Since inflammation is central to candidiasis, we aimed to define the role of p38γ/p38δ in the host defence against disseminated candidiasis as mirrored by an intravenous challenge model of *C. albicans* infection. This is an established model of systemic candidiasis, in which the kidneys are the primary target organs; mice develop renal failure and septic shock, and this recapitulates the progressive sepsis seen in humans during severe clinical cases (Spellberg *et al*, 2005). We first compared the survival of p38γ/

p38δ-deficient (p38γ/δ$^{-/-}$) mice and control WT mice to fungal infection. We observed that the lack of p38γ/p38δ caused a remarkable protection to infection (Fig 3A). To determine whether the effect in survival was due to p38γ/p38δ in myeloid cells, we also analysed mice with myeloid cell-specific p38γ/p38δ deletion (LysM-p38γ/δ$^{-/-}$; Zur *et al*, 2015). In agreement with the observation in p38γ/δ$^{-/-}$ mice, the survival of LysM-p38γ/δ$^{-/-}$ mice was significantly increased compared with control mice after *C. albicans* infection (Fig 3A). The effect of p38γ/p38δ deletion on mouse survival was probably due to the decreased load of *C. albicans* in the organs; accordingly, the fungal burden in p38γ/δ$^{-/-}$ and LysM-p38γ/δ$^{-/-}$ kidneys was significantly lower than in WT control mice (Fig 3B and Appendix Fig S3). When we analysed *C. albicans* dissemination after intravenous injection, we found that the fungus was rapidly cleared from the bloodstream in all genotypes (Appendix Fig S3). Although the clearance in WT mice bloodstream was markedly slower than in p38γ/δ$^{-/-}$ and LysM-p38γ/δ$^{-/-}$ mice, the fungal burden in spleen, liver and brain was similar in all mice (Appendix Fig S3). In kidney, fungal burden was similar in all mice at day 1, whereas in p38γ/δ$^{-/-}$ and LysM-p38γ/δ$^{-/-}$ mice, the fungal burden was slightly lower at day 2 and significantly lower at day 3 of *C. albicans* injection than in WT mice (Appendix Fig S3). Histological analysis of kidney sections revealed that at day 3 post-infection WT mice showed evident growth of *C. albicans* forming hyphae, whereas kidneys from p38γ/δ$^{-/-}$ and LysM-p38γ/δ$^{-/-}$ mice displayed nearly undetectable hyphae formation (Fig 3C). These results indicate that p38γ/p38δ, particularly in myeloid cells, increase resistance to *C. albicans* infection.

Since p38γ/p38δ regulate TPL2 protein levels in macrophages, we then assessed the role of TPL2 in the response to *C. albicans in vivo*. Loss of TPL2 did not protect against fungal infection, as TPL2$^{+/+}$ and TPL2$^{-/-}$ mice survival was similar (Fig 3D). We also found that TPL2 deletion led to a small increase in fungal burden in the kidney, when compared with TPL2$^{+/+}$ kidney (Fig 3E). Histological analysis of kidney sections confirmed these results; both WT and TPL2$^{-/-}$ mice showed extensive growth of *C. albicans* forming hyphae (Fig 3C). These data show that TPL2 deletion does not protect against *C. albicans* infection and suggest that *in vivo* p38γ/p38δ diminish fungal infection independently of TPL2.

## p38γ/p38δ deletion decreased the inflammatory response against *Candida albicans* infection

To address the possibility that increased survival of p38γ/δ-deficient mice to *C. albicans* infection could be due to effects on the inflammatory response, we quantified the levels of pro-inflammatory cytokines at early time points of infection. p38γ/δ$^{-/-}$ mice showed a significant reduction of TNFα and IFNγ serum levels compared to WT mice (Appendix Fig S4A). IL-1β production was slightly reduced in p38γ/δ$^{-/-}$ mice compared to control (Appendix Fig S4A). Accordingly, in kidney, *TNFα*, *IL6* and *IL-1β* mRNA levels were significantly reduced in p38γ/δ$^{-/-}$ and LysM-p38γ/δ$^{-/-}$ mice at day 1 post-infection, compared to WT mice (Fig 4A). The reduction in cytokine mRNA production was more pronounced in LysM-p38γ/δ$^{-/-}$ than in p38γ/δ$^{-/-}$ mice. In TPL2$^{-/-}$ mice infected with *C. albicans,* the production of cytokines in the

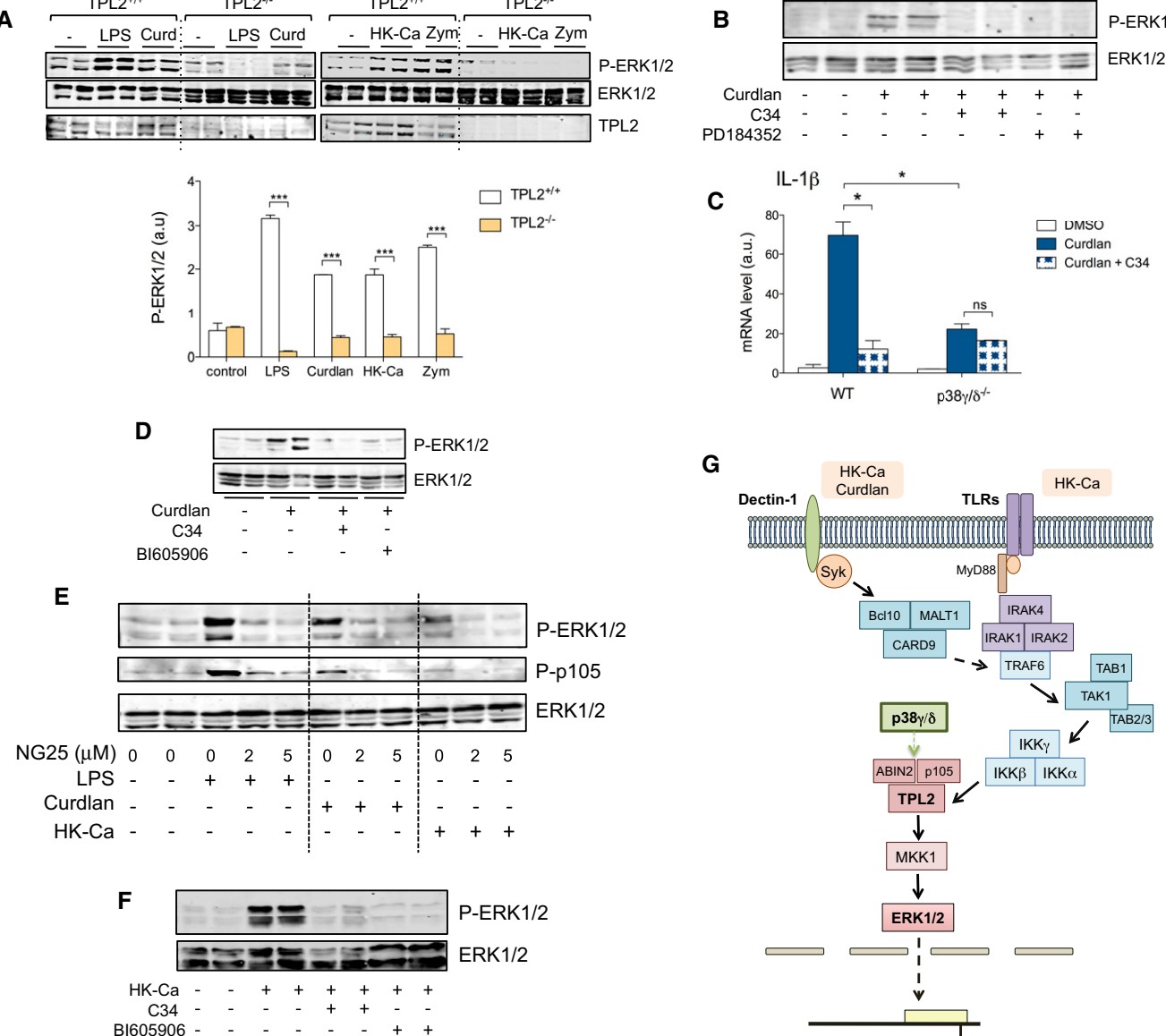

**Figure 2. ERK1/2 activation is mediated by TPL2 in Dectin-1 signalling.**

A   BMDM from TPL2[+/+] or TPL2[−/−] mice were stimulated with 1 × 10[6] CFU/ml HK-Ca, 10 μg/ml Curdlan or 50 μg/ml Zymosan for 1 h, or with 100 ng/ml LPS for 30 min. Cell lysates were immunoblotted with the indicated antibodies. Representative immunoblots are shown. Bands from three experiments were quantified using the Odyssey infrared imaging system (bottom), and data show mean ± SEM from two experiments in duplicate. ***$P \leq 0.001$. Parametric, unpaired *t*-test.

B   WT BMDM were incubated for 1 h with or without 5 μM C34 or 2 μM PD184352, and then stimulated for 1 h with 10 μg/ml Curdlan. Representative immunoblots from two independent experiments are shown.

C   BMDM were incubated for 1 h with DMSO or with 5 μM C34, and then exposed for 1 h to 10 μg/ml Curdlan. Relative mRNA expression was determined by qPCR for IL-1β. Results were normalized and fold induction calculated. Data show mean ± SEM from one representative experiment of two in triplicate, with similar results. ns, not significant, *$P \leq 0.05$ relative to WT BMDM exposed to Curdlan. Parametric, unpaired *t*-test.

D   WT BMDM were incubated for 1 h with or without 5 μM C34 or 10 μM BI605906 and then stimulated with Curdlan as in (B). Representative immunoblots from four independent experiments are shown.

E   WT BMDM were incubated for 1 h with DMSO or with 2 or 5 μM NG25 and then stimulated with HK-Ca, Curdlan or LPS as in (A). Representative immunoblots from two independent experiments are shown.

F   WT BMDM were incubated for 1 h in the absence or the presence of 5 μM C34 or 10 μM BI605906, and then stimulated for 1 h with 1 × 10[6] CFU/ml HK-Ca. Cell lysates were immunoblotted as indicated. Representative blots from two independent experiments are shown.

G   Schematic representation of the Dectin-1 signalling pathways involved in ERK1/2 activation, which is controlled by TAK1-IKK-TPL2 in HK-Ca and Curdlan-stimulated macrophages. The activation of the TAK1 complex (TAB 1-TAK1-TAB 2/3) and of the IKK pathway (Cohen, 2014) might be mediated by TRAF6, which binds to the CARD9/BCL-10/MALT1 complex downstream of Syk (Geijtenbeek & Gringhuis, 2009). TLR stimulation by HK-Ca also triggers the activation of TAK1-IKK-TPL2 via MyD88. p38γ and p38δ regulate TPL2 steady-state levels, which is in a complex with ABIN-2 and p105 (Gantke *et al*, 2011).

Source data are available online for this figure.

kidney was not reduced compared to TPL2$^{+/+}$ mice (Appendix Fig S4B). *IL6*, *TNFα*, and *IL-1β* mRNA levels in TPL2$^{-/-}$ kidney were significantly higher than in TPL2$^{+/+}$ mice (Appendix Fig S4B) indicating that *in vivo* TPL2 does not mediate cytokine production downstream of p38γ/p38δ in response to *C. albicans*.

We also analysed, by flow cytometry, different leucocyte cell types infiltrating the infected kidney of WT, p38γ/δ$^{-/-}$ and LysM-p38γ/δ$^{-/-}$ mice, at days 0, 1 and 3 after *C. albicans* infection. The number of total renal-infiltrating leucocytes (CD45$^+$ cells) increased with the infection and was significantly lower in p38γ/δ$^{-/-}$ and LysM-p38γ/δ$^{-/-}$ mice than in WT at day 3 post-infection (Fig 4B). Next, we determined the recruitment of major leucocyte types involved in inflammation: macrophages, neutrophils and T cells. We found that *C. albicans* infection caused an increase in the amount of F4/80$^+$ macrophages and Ly6G$^+$ neutrophils in all genotypes after infection (Fig 4B). Macrophages and neutrophils are a major fraction of the total CD45$^+$ population after *C. albicans* infection (Fig 4B). The amount of macrophages and neutrophils accumulated in p38γ/δ$^{-/-}$ and LysM-p38γ/δ$^{-/-}$ kidney 3 days after *C. albicans* infection was significantly lower than in WT, although the effect in neutrophil recruitment was more pronounced than in macrophages recruitment (Fig 4B). The reduction in macrophages and neutrophils recruitment was more evident in LysM-p38γ/δ$^{-/-}$ than in p38γ/δ$^{-/-}$ mice. CD4$^+$, but not CD8$^+$, T lymphocyte amount increased after *C. albicans* infection; however, they remain a minor percentage of the total leucocyte population and the difference in T-cell accumulation was minimal between WT, p38γ/δ$^{-/-}$ and LysM-p38γ/δ$^{-/-}$ mice (Appendix Fig S4C). We further confirmed the decrease in F4/80$^+$ macrophages and Ly6G$^+$ neutrophils recruitment using an intraperitoneal *C. albicans* model (Netea *et al*, 1999). As expected, we found a significant reduction in peritoneal neutrophil recruitment in p38γ/δ$^{-/-}$ and LysM-p38γ/δ$^{-/-}$ mice compared to control; however, the recruitment of F4/80$^+$ macrophages was clearly reduced only in LysM-p38γ/δ$^{-/-}$ peritoneum (Fig 4C).

The decrease in leucocyte recruitment in p38γ/δ$^{-/-}$ and LysM-p38γ/δ$^{-/-}$ kidneys suggests reduced chemokine expression. Thus, *CCL2*, *MIP-2* and *KC* mRNA levels were clearly lower in the kidney of p38γ/δ$^{-/-}$ and LysM-p38γ/δ$^{-/-}$ mice than in WT animals at day 1 post-infection (Fig 4D). Together, these findings suggest that p38γ/p38δ are involved in the early inflammatory response to *C. albicans* by modulating cytokine and chemokine production and the recruitment of leucocytes into the *C. albicans*-infected kidney. We then treated WT and p38γ/δ$^{-/-}$ mice with ibuprofen, which is a commonly used antiinflammatory compound (Vilaplana *et al*, 2013). Ibuprofen treatment reduced kidney *C. albicans* load and neutrophil recruitment in WT mice to similar levels than the loss of

p38γ/p38δ (Appendix Fig S5A and B). In p38γ/δ$^{-/-}$ mice, the treatment with ibuprofen did not cause a major effect in fungal burden or the recruitment of neutrophils after *C. albicans* infection (Appendix Fig S5A and B). When we examined the effect of ibuprofen on the survival of *C. albicans*-infected WT and p38γ/δ$^{-/-}$ mice, we found that at early times the antiinflammatory compound caused a marked protection and an increase in the survival of WT mice (Appendix Fig S5C). Ibuprofen, however, did not affect p38γ/δ$^{-/-}$ mice survival (Appendix Fig S5C). These results support that p38γ/p38δ are involved in modulating an early deleterious inflammatory response to *C. albicans*.

## p38γ/p38δ deficiency increases macrophages and neutrophils antifungal activity

Macrophages and neutrophils regulate bacterial and fungal infections by phagocytosis and killing mechanisms (Vonk *et al*, 2002; Nicola *et al*, 2008; Lionakis *et al*, 2013). The production of highly reactive nitrogen and oxygen species (RNS and ROS) is one of the main mechanisms used by phagocytes to control fungal infection and is essential for *C. albicans* killing (Nicola *et al*, 2008; Naglik *et al*, 2014). Our data suggest that *C. albicans* can be eliminated more efficiently in p38γ/δ-null mice than in WT controls. We then analysed whether or not the p38γ/p38δ deletion affected the ability of macrophages to express inducible nitric oxide synthase (iNOS), which generates nitric oxide (NO) from arginine and oxygen and is expressed after the activation of phagocytic cells. *iNOS* mRNA expression was upregulated upon HK-Ca stimulation in both WT and p38γ/δ$^{-/-}$ BMDM; however, the levels in p38γ/δ$^{-/-}$ cells were markedly higher than in WT (Fig 5A). We also found that the loss of p38γ/p38δ significantly enhanced the production of reactive oxygen species (ROS) by *C. albicans*-stimulated macrophages (Fig 5B). We then performed an *ex vivo* killing assay, co-culturing *C. albicans* with BMDM, to assess whether the enhanced iNOS and ROS levels in p38γ/δ$^{-/-}$ cells correlated with an increase in their *Candida*-killing potency. Fungal phagocytosis by BMDM was similar in both genotypes (Appendix Fig S6A), indicating that the recognition of *C. albicans* is not affected in p38γ/δ$^{-/-}$ cells. Nonetheless, p38γ/δ$^{-/-}$ macrophages were significantly more efficient in the killing of live *C. albicans* than WT cells (Fig 5C). The ROS production and the fungal-killing capacity of p38γ/δ$^{-/-}$ neutrophils were also markedly higher compared to WT control neutrophils (Fig 5D and E), further indicating that p38γ/p38δ negatively impact the ability of phagocytic cells to clear the *C. albicans*, which is important for mice survival in the candidiasis model.

To examine whether the observed differences in ROS production *in vitro* were also seen *in vivo* in immune cells that are recruited to

---

**Figure 3. p38γ/p38δ deletion decreases *Candida albicans* infection in mice.**

A, B   WT, p38γ/δ$^{-/-}$ and LysM-p38γ/δ$^{-/-}$ mice were infected with $1 \times 10^5$ CFU *C. albicans*. (A) Survival monitored as indicated. Data are presented as a Kaplan–Meier plot from two independent experiments ($n$ = 20 mice per genotype). Two-way ANOVA using GraphPad Prism software. (B) Kidney fungal burden at day 3 post-infection. Data are expressed as CFU/g kidney (mean ± SEM). Each symbol represents an individual mouse. ns, not significant, *$P \leq 0.05$, **$P \leq 0.01$ and ***$P \leq 0.001$ relative to WT mice. Parametric, unpaired *t*-test.

C       Representative PAS-haematoxylin staining of kidney sections from mice at day 3 post-infection. Bottom panels are high magnification of the area marked by a dotted square in the top panels. Scale bars are 100 μm.

D, E   TPL2$^{+/+}$ and TPL2$^{-/-}$ mice were infected with *C. albicans* as in (A). (D) Death was monitored. Data are a summary of two independent experiments ($n$ = 12 mice per genotype). (E) Kidney fungal burden at day 3 post-infection with $1 \times 10^5$ CFU. Each symbol represents an individual mouse. Data are expressed as CFU/g kidney (mean ± SEM). *$P \leq 0.05$ relative to WT mice. Two-way ANOVA using GraphPad Prism software.

    

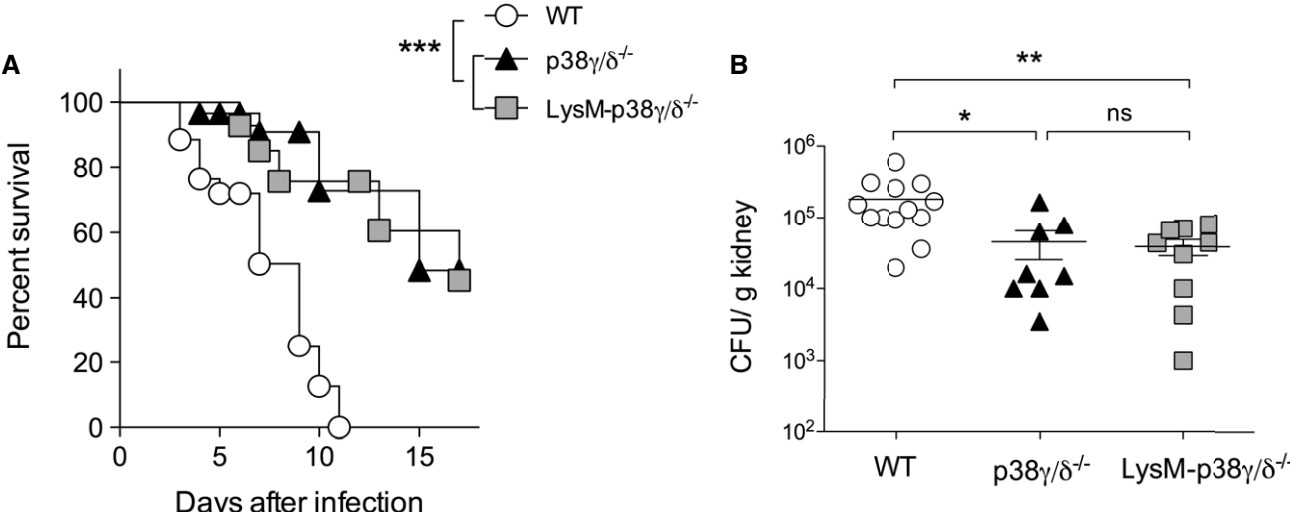

Figure 3.

the inflammatory sites, we infected intraperitoneally WT and p38γ/δ$^{-/-}$ mice with *C. albicans* and 24 h later measured ROS production both without (Fig 5F) or with (Fig 5G) re-stimulation *in vitro* with *C. albicans*. Cells that infiltrate the intraperitoneal cavity of p38γ/δ$^{-/-}$ mice produced significantly more ROS than cells from WT mice (Fig 5F and G). Additionally, the induction of *iNOS* mRNA production in the kidney of *C. albicans*-infected mice was markedly higher in p38γ/δ$^{-/-}$ than in WT mice (Fig 5H). All these data suggest that the protective effect of p38γ/δ deficiency against fungal infection is due to increased killing of *C. albicans*.

To investigate whether the increased levels of ROS observed in p38γ/δ$^{-/-}$ BMDM contribute to antifungal activity of these cells against *C. albicans*, we treated them with the antioxidant compound N-acetylcysteine (NAC; Victor *et al*, 2003). NAC reduced both ROS levels and *C. albicans* killing *in vitro* (Appendix Fig S6B and C). Moreover, we found that *in vivo* the antioxidant compound decreased the survival of p38γ/δ$^{-/-}$-infected mice to similar levels than those in WT-infected mice (Fig 5I). These results indicate that the increase in ROS production observed in p38γ/δ$^{-/-}$ mice is important for the antifungal activity and protection against *C. albicans* infection.

**Pharmacological inhibition of p38γ/p38δ ameliorates *Candida albicans* infection**

Based on our data, we hypothesized that p38γ/p38δ inhibition *in vivo* might improve the outcome of *C. albicans* infection and provide evidence for a new therapeutic approach. We performed experiments treating mice with the p38MAPK inhibitor BIRB796. Since BIRB796 inhibits all p38 isoforms (Kuma *et al*, 2005), we also used as control the compound SB203580, which only blocks p38α/p38β activity (Kuma *et al*, 2005; Bain *et al*, 2007), so we could determine which effect was caused by p38γ/p38δ or by p38α/p38β inhibition. We first checked whether these compounds inhibit p38MAPKs in the kidney of infected mice. Both compounds worked equally well *in vivo* as shown by the reduction in *TNFα* mRNA expression in the kidney of *C. albicans*-infected mice (Appendix Fig S7A) and inhibited p38α, as shown by the loss of its phosphorylation (Appendix Fig S7B). Moreover, BIRB796, but not SB203580, was able to decrease the phosphorylation of p38γ and p38δ induced after *C. albicans* infection (Appendix Fig S7C). Notably, BIRB796 treatment significantly reduced kidney fungal load at day 3 (Fig 6A), but not at day 1 (Appendix Fig S7D) in WT mice. This fungal burden reduction was similar to that observed in p38γ/δ$^{-/-}$ mice (Appendix Fig S7E). BIRB796 treatment did not affect *C. albicans* load in p38γ/δ$^{-/-}$ mice (Appendix Fig S7E). SB2013580 did not affect *C. albicans* growth in the kidney compared to control mice treated with the vehicle DMSO (Fig 6A). Consequently, BIRB796 treatment led to a higher fungal-induced *iNOS* mRNA levels compared to controls (Fig 6B). Treatment with the inhibitor also decreased the recruitment of neutrophils to the kidney (Fig 6C). We confirmed this using the intraperitoneal *C. albicans* infection model and found that only the treatment with BIRB796 led to a significant reduction in peritoneal neutrophil recruitment compared to control mice treated with either DMSO or SB203580 (Fig 6D). Thus, p38γ/p38δ inhibition enhances antifungal immunity and might protect from *C. albicans* sepsis.

## Discussion

In this study, we address the role of p38γ/p38δ in candidiasis and show their importance in the regulation of innate antifungal immunity. We found that deletion of p38γ/p38δ has a beneficial effect improving survival in *C. albicans*-infected mice, and that p38γ/p38δ deficiency in myeloid cells is fundamental for this phenotype.

Macrophages influence the inflammatory environment by modulating the production of cytokines and chemokines during fungal infection (Ersland *et al*, 2010), and p38γ/p38δ deletion limited the intrinsic response of BMDM to *C. albicans* by diminishing the production of cytokines and chemokines. Upon fungus recognition, the stimulation of macrophage receptors triggers the activation of signalling pathways, such as the ERK1/2 pathway, which induce the expression of immune modulators. Here, we show that, in addition to their role in TLR signalling, TAK1-IKKβ-TPL2 contribute to Dectin-1-mediated signalling in mouse BMDM, being essential for ERK1/2 activation. Upon Dectin-1 stimulation, the protein Syk is recruited to the receptor, which triggers the assembly of the complex CARD9/BCL-10/MALT1 (CMB; Geijtenbeek & Gringhuis, 2009). It has been shown that the CBM complex activates the IKK pathway by recruiting and activating the protein E3 ubiquitin ligase TNF receptor-associated factor 6 (TRAF6), which is essential for TAK1 activation (Geijtenbeek & Gringhuis, 2009; Cohen, 2014). We therefore propose the existence of a Syk-CMB-TRAF6-TAK1-IKKβ-TPL2-MKK1-ERK1/2 pathway in macrophages, which is activated by Dectin-1 engagement and regulated by p38γ/p38δ (Fig 2G).

Contrary to what happens in p38γ/p38δ-deficient mice, the lack of TPL2 does not affect the survival of *C. albicans*-infected mice, nor reduce kidney fungal load or cytokine production, suggesting that p38γ/p38δ signalling controls systemic candidiasis *in vivo* independently of TPL2. This result was unexpected since in macrophages the combined deletion of p38γ and p38δ decreases the steady-state levels of TPL2 and therefore the activation of ERK1/2 downstream of TLR and Dectin-1, which has a central role in cytokine production (Risco *et al*, 2012; Arthur & Ley, 2013). Nonetheless, this finding is supported by previous reports in which p38γ/p38δ deletion does not recapitulate the effect observed in TPL2$^{-/-}$ mice. In an azoxymethane/dextran sodium sulphate (AOM/DSS) colitis-associated colon cancer model, TPL2 deficiency increases the development of tumours in mice compared to WT mice, whereas in p38γ/δ$^{-/-}$ mice tumour development is decreased (Koliaraki *et al*, 2012; Del Reino *et al*, 2014). The role of TPL2 in candidiasis has not been addressed so far, and further studies using TPL2-deficient mice are required to determine its specific function in fungal infection. Additionally, it is possible that p38γ/p38δ, and also TPL2, have different or even opposite functions depending on the cell type or context, which would affect the general outcome in the response to *C. albicans* infection. This idea is supported by the observation that: (i) the reduction in both cytokine production and leucocyte recruitment is significantly more pronounced in LysM-p38γ/δ$^{-/-}$ than in p38γ/δ$^{-/-}$ mice; (ii) cytokine production is reduced by pharmacological TPL2 inhibition in macrophages but not in TPL2-deficient mice infected with *C. albicans*; and (iii) the production of TNFα and IL-6 is reduced in p38γ/δ$^{-/-}$-infected mice, but not in p38γ/δ$^{-/-}$-infected macrophages, suggesting that the expression levels of these molecules in infected mice are indirectly regulated by p38γ/δ in myeloid cells acting on other cell types.

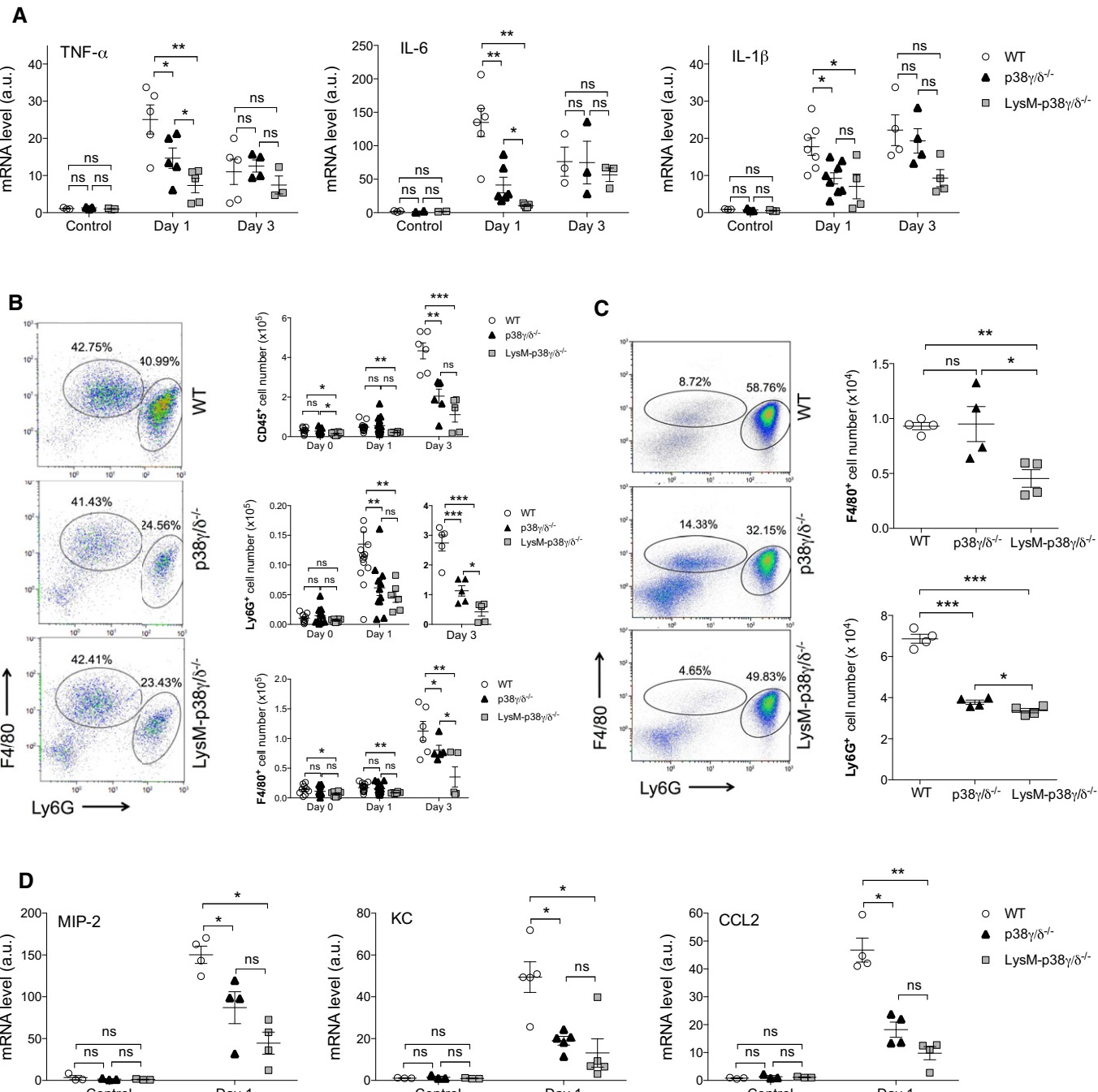

**Figure 4.  Reduced inflammation in p38γ/δ$^{-/-}$ mice in response to *Candida albicans* infection.**

A  WT, p38γ/δ$^{-/-}$ and LysM-p38γ/δ$^{-/-}$ mice were intravenously infected with $1 \times 10^5$ CFU *C. albicans* and at days 1 and 3 post-infection, relative TNFα, IL-6 and IL-1β mRNA expression in the kidney was determined by qPCR and normalized to β-actin mRNA. Each symbol represents an individual mouse. Figure shows mean ± SEM (n = 5–8). ns, not significant, *$P \leq 0.05$, **$P \leq 0.01$ relative to WT mice. Parametric, unpaired *t*-test.

B  Kidney cells from 0-, 1- and 3-day *C. albicans*-treated WT, p38γ/δ$^{-/-}$ and LysM-p38γ/δ$^{-/-}$ mice as in (A) were stained with anti-CD45, -Ly6G and -F4/80 antibodies and positive cells analysed by flow cytometry. CD45$^+$ cells were gated and -F4/80$^+$ and -Ly6G$^+$ cells analysed by flow cytometry. Representative profiles are shown. Each symbol represents an individual mouse (two to three independent experiment). Figure shows mean ± SEM (n = 5–14 mice/condition), ns, not significant; *$P \leq 0.05$, **$P \leq 0.01$, ***$P \leq 0.001$, relative to WT kidney cells, at each time point. Parametric, unpaired *t*-test.

C  Mice were intraperitoneally infected with $5 \times 10^6$ CFU *C. albicans*, and at day 1 post-infection, peritoneal cells were stained and analysed as in (B). Representative profiles are shown. Each symbol represents an individual mouse. Figure shows mean ± SEM (n = 4), ns not significant; *$P \leq 0.05$, **$P \leq 0.01$, ***$P \leq 0.001$, relative to WT kidney cells. Parametric, unpaired *t*-test.

D  Mice were infected with *C. albicans* as in (A), relative MIP-2, KC and CCL2 mRNA expression in the kidney was determined by qPCR as in (A). Each symbol represents an individual mouse. Figure shows mean ± SEM (n = 3–5). ns, not significant, *$P \leq 0.05$, **$P \leq 0.01$ relative to WT mice. Parametric, unpaired *t*-test.

    

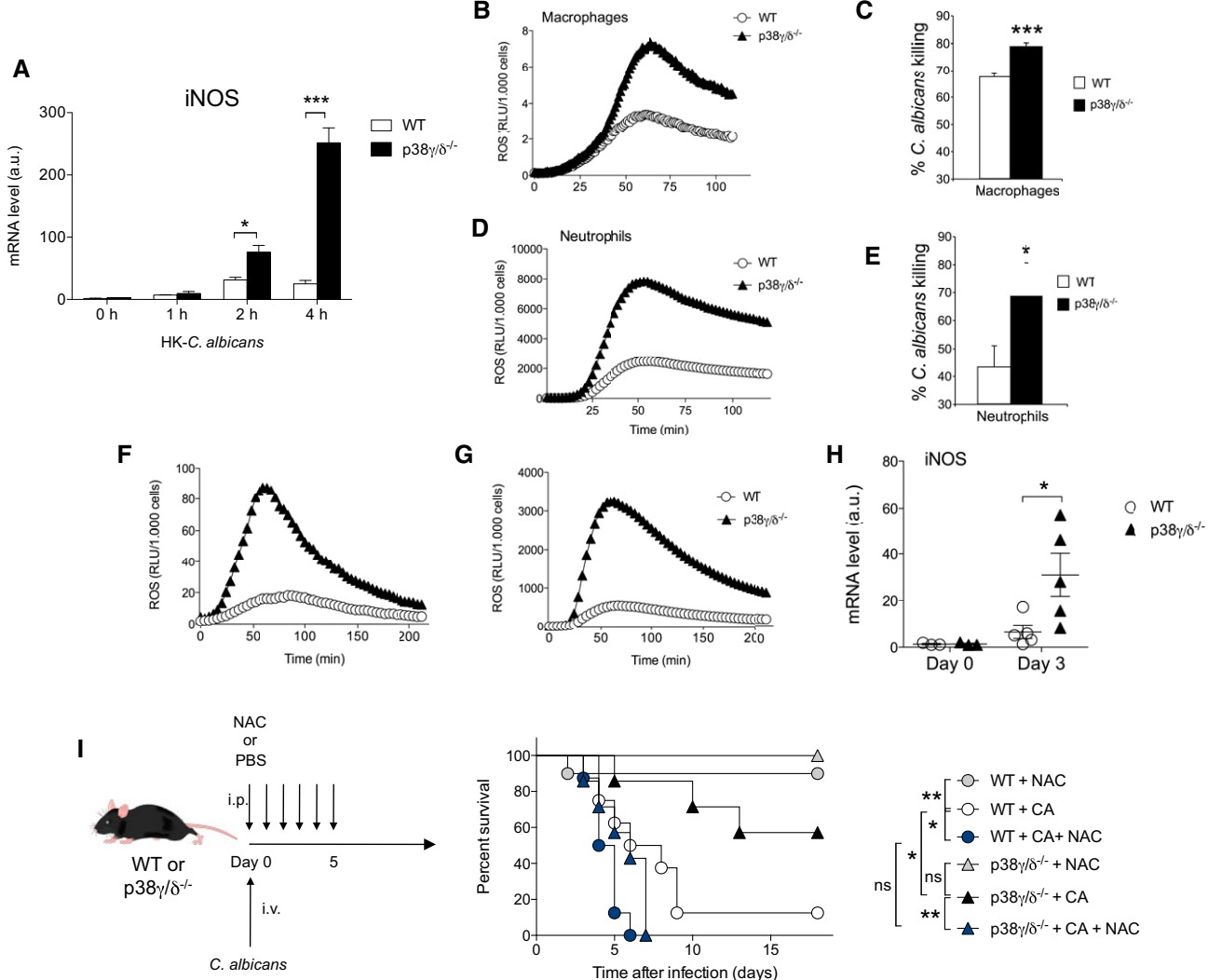

**Figure 5.  p38γ/p38δ regulate antifungal activity against *Candida albicans*.**

A    BMDM from WT or p38γ/δ$^{-/-}$ mice were exposed for the indicated times to $1 \times 10^6$ CFU/ml HK-Ca. Relative iNOS mRNA expression was determined by qPCR. Data show mean ± SEM from one representative experiment of two in triplicate, with similar results. *$P \leq 0.05$, ***$P \leq 0.001$ relative to WT BMDM. Parametric, unpaired *t*-test.

B    ROS production in BMDM co-cultured with $1 \times 10^6$ CFU *C. albicans*. Experiments were performed in triplicate. RLU, relative light unit.

C    *Candida albicans* killing by BMDM. The results are represented as percentage of killing (the percentage of killed fungal cells among the phagocytosed fungus). The *C. albicans*/BMDM ratio was 1:10. Values are mean ± SEM ($n = 6$); ***$P \leq 0.001$ relative to WT cells. Parametric, unpaired *t*-test.

D    ROS production in neutrophils isolated from blood of WT or p38γ/δ$^{-/-}$ mice after infection. Neutrophils were stimulated with $1.5 \times 10^6$/ml HK-Ca. Experiments were performed in triplicate.

E    Candicidal activity of neutrophils determined as in (C). Values are mean ± SEM ($n = 4$); *$P \leq 0.05$ relative to WT cells. Parametric, unpaired *t*-test.

F, G    ROS production without (F) or with (G) re-stimulation with *C. albicans* in intraperitoneal immune cell infiltrates of WT or p38γ/δ$^{-/-}$ mice 1 day after intraperitoneal *C. albicans* infection ($5 \times 10^6$ CFU, $n = 4$ mice per group).

H    WT and p38γ/δ$^{-/-}$ mice were intravenously infected with $1 \times 10^5$ CFU *C. albicans*. Relative iNOS mRNA expression in the kidney was determined by qPCR and normalized to β-actin mRNA. Each symbol represents an individual mouse. Figure shows mean ± SEM ($n = 3$–5). Only significant results are indicated, *$P \leq 0.05$, relative to WT mice. Parametric, unpaired *t*-test.

I    WT and p38γ/δ$^{-/-}$ mice were infected with $1 \times 10^5$ CFU *C. albicans* and treated with 200 mg/kg body weight per day of N-acetylcysteine (NAC) from SIGMA or with the same volume of the vehicle PBS for 5 days [WT + CA + NAC ($n = 8$); p38γ/δ$^{-/-}$ + CA + NAC ($n = 8$)]. Control groups of WT and p38γ/δ$^{-/-}$ mice treated with 200 mg/kg body weight per day of NAC were included to check its toxicity [WT + NAC ($n = 10$); p38γ/δ$^{-/-}$ + NAC ($n = 10$)]. Control groups of WT and p38γ/δ$^{-/-}$ mice infected with *C. albicans* were also included for comparison [WT + CA ($n = 8$); p38γ/δ$^{-/-}$ + CA ($n = 8$)]. Survival was monitored as indicated. Data are presented as a Kaplan–Meier plot. ns, not significant, *$P \leq 0.05$; **$P \leq 0.01$. Two-way ANOVA using GraphPad Prism software.

The lower susceptibility of p38γ/δ$^{-/-}$ and LysM-p38γ/δ$^{-/-}$ mice to systemic infection with *C. albicans* is likely due to the decreased fungal growth in their kidneys. The kidney fungal load correlates with the severity of renal failure and the associated progressive sepsis, which finally is the cause of death in this model (Spellberg *et al*, 2005). One important finding of the present study is that the

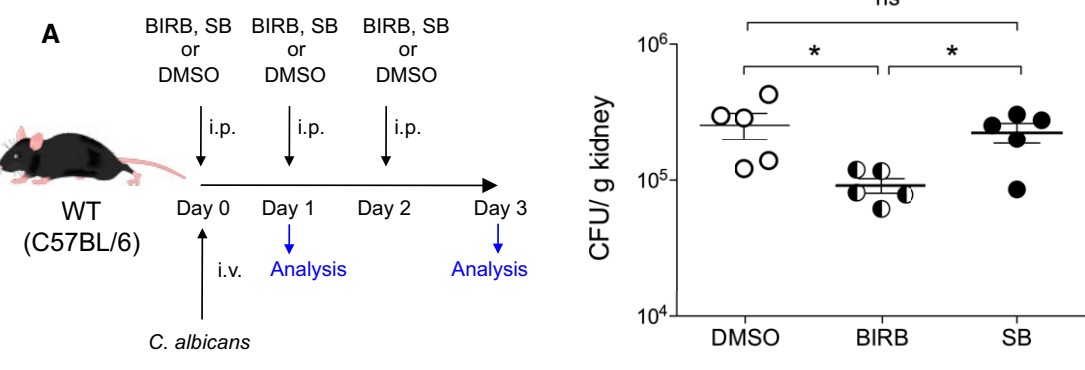

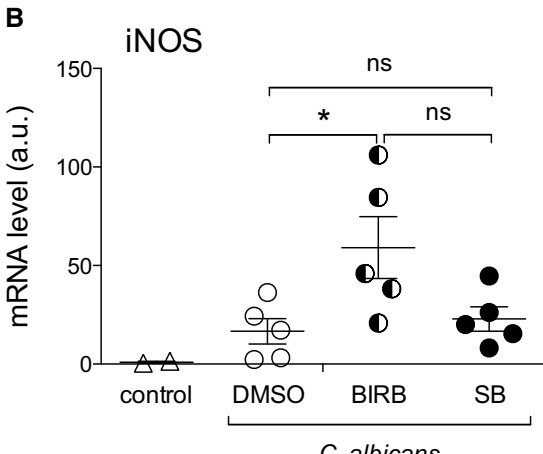

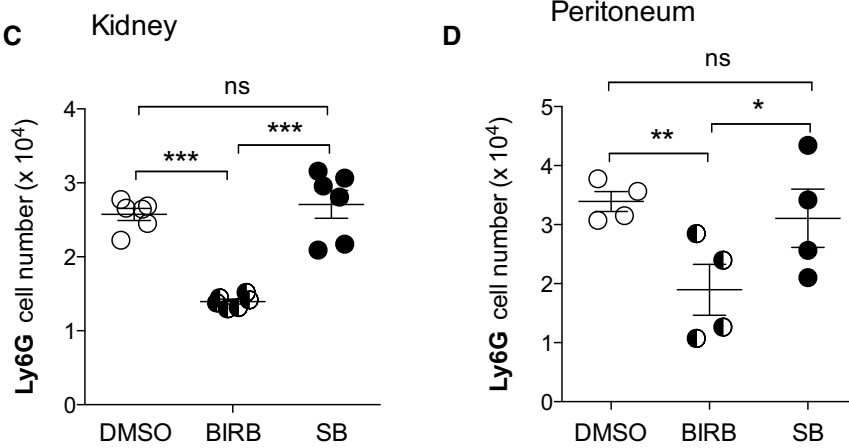

**Figure 6.  Treatment with p38γ/p38δ inhibitor shows antifungal effects *in vivo*.**

A    WT mice were intravenously injected with $1 \times 10^5$ CFU of *Candida albicans* and treated with 10 mg BIRB796 or SB203580 per kg body weight per day, or with the same volume of the vehicle DMSO. Kidney fungal load was determined 3 days after infection. ns, not significant, *$P \leq 0.05$ ($n$ = 5 mice/condition). Each symbol represents an individual mouse. Parametric, unpaired $t$-test.

B    Mice were treated as in (A) and iNOS mRNA levels in the kidney measured 3 days after infection by qPCR. Each symbol represents an individual mouse. Figure shows mean ± SEM ($n$ = 5 mice/condition). ns, not significant, *$P \leq 0.05$. Parametric, unpaired $t$-test.

C    Neutrophil infiltration in the kidney of infected WT mice, treated with BIRB796 or SB203580 inhibitor as in (A) was determined by flow cytometry. Each symbol represents an individual mouse. Figure shows mean ± SEM ($n$ = 6 mice/condition), ns not significant; ***$P \leq 0.001$. Parametric, unpaired $t$-test.

D    Mice were intraperitoneally infected with $5 \times 10^6$ CFU *C. albicans* and treated with 10 mg/kg body weight per day BIRB796 or SB203580, or with the same volume of the vehicle DMSO. Neutrophil infiltration in the peritoneum was measured by flow cytometry at day 1 post-infection. Each symbol represents an individual mouse. Figure shows mean ± SEM ($n$ = 4 mice/condition), ns, not significant; *$P \leq 0.05$, **$P \leq 0.01$. Parametric, unpaired $t$-test.

production of cytokines such as IL-6, TNFα, IFNγ, and IL-1β, in p38γ/δ−deficient mice is significantly lower than in WT animals in response to *C. albicans*. p38γ/p38δ deletion also impaired the production of the neutrophil and macrophage chemoattractants CCL2, KC and MIP-2 (Soehnlein & Lindbom, 2010), which paralleled the decrease in leucocyte recruitment to the infected kidneys, particularly of neutrophils. It has been shown that pharmacological suppression of monocytes and neutrophils with the compound pioglitazone, a nuclear receptor peroxisome proliferator-activated receptor-γ (PPAR-γ) agonist, led to increased survival and reduced immunopathology of *C. albicans*-infected mice (Majer *et al*, 2012), in agreement with our results. Thus, the decrease in inflammatory molecule production observed in p38γ/p38δ-deficient mice might contribute to the increase in mouse survival as our results from the treatment with the antiinflammatory drug ibuprofen indicate. Perhaps the reduced acute inflammatory response in p38γ/δ$^{-/-}$ mice is more successful in allowing the elimination of the fungus and preventing kidney damage. However, alternative explanations could be considered. One such explanation may envisage the lower inflammation to be due to the lower fungal loads in the kidneys as a result of the successful early elimination of the pathogen. Effects of p38γ/δ on inhibitory cell populations with deleterious roles during candidiasis, such as T-regulatory cells (Netea *et al*, 2008), can also be considered since p38γ/p38δ deletion slightly decreased the recruitment of CD4$^+$ cells in the kidney of infected mice. The potential effects of p38γ/δ in the activation of T-regulatory cells are unknown and require investigation in future studies.

The inhibitory effects of p38γ/p38δ on myeloid cells killing mechanisms, such as ROS production or iNOS expression, might also contribute to the protection to *C. albicans* infection observed in p38γ/p38δ-deficient mice. Macrophages and neutrophils are important not only for building an inflammatory environment, but also for the fungal clearance to achieve an efficient protection against systemic candidiasis (Netea *et al*, 2008). A balance between the beneficial immune response that kills the pathogen and an exacerbated inflammation with negative effects is essential for the resolution of the infection without causing tissue damage in the host. Our data show that at an early phase (day 1) of *C. albicans* infection the loss of p38γ/p38δ does not affect fungal burden in the kidney, whereas, at a later phase of infection (day 3) fungal burden is lower in p38γ/p38δ-deficient mice than in control WT. These results indicate that p38γ/p38δ deletion favours fungal clearance in systemic candidiasis, which correlates with the increased killing capacity, ROS production and *iNOS* mRNA levels observed in p38γ/p38δ-null cells and mice compared to control WT. Accordingly, a decrease in ROS levels caused by the treatment with the antioxidant agent NAC correlates with a decrease in *C. albicans* killing by macrophages *in vitro* and a reduction in mice survival *in vivo*. These observations suggest that the lack of p38γ/p38δ facilitates fungus elimination by phagocytic cells, which would control fungal growth in the kidney, limiting the inflammatory immune response and improving mouse survival after fungal infection.

The *in vivo* inhibition of p38γ/p38δ with the compound BIRB796 mimicked the loss of these kinases, with regard to fungal burden and iNOS production in the kidney, as well as to the recruitment of neutrophils in the kidneys and the peritoneum after *C. albicans* infection. This reveals the potential value of p38γ/p38δ inhibitors as effective antifungal agents and also indicates the importance of the development of potent and specific p38γ/p38δ inhibitors as an alternative to traditional p38α inhibitors. In prolonged treatments, p38α inhibitors have proven to be minimally effective due to liver toxicity or to increased inflammation due to the failure of p38α-mediated negative regulation of TAK1 (Gaestel *et al*, 2009).

In summary, our work adds knowledge into the understanding of the molecular mechanisms responsible for the host defence against *C. albicans* infection and provides evidence that p38γ/p38δ play an important role in the control of fungal infection at two different but interconnected levels: the potentially harmful host acute inflammatory response and the beneficial host immune response that eliminates the *Candida*. These two processes modulate each other. Considering the complexity of C-type lectin receptor signalling and that the incidence of *Candida species* infections and the relapse episodes after antifungal treatment have increased in recent decades (Brown *et al*, 2012), further studies of p38γ/p38δ-associated signalling pathways would offer novel strategies for the design of more effective agents against fungal infections.

# Materials and Methods

### Mice

All mice (13–18 weeks old female) were housed in specific pathogen-free conditions in the CNB-CSIC animal house, and all animal procedures were performed in accordance with national and EU guidelines, with the approval of the Centro Nacional de Biotecnología Animal Ethics Committee, CSIC and Comunidad de Madrid (Reference: CAM PROEX 316/15).

C57BL/6J TPL2$^{+/+}$ and C57BL/6J TPL2$^{-/-}$ littermates were produced from heterozygous mice (Rodriguez *et al*, 2008). C57BL/6J WT, p38γ/δ$^{flox/flox}$, p38γ/δ$^{-/-}$ and LysM-Cre$^{+/-}$-p38γ/δ$^{flox/flox}$ (called here LysM-Crep38γ/δ$^{-/-}$) mice have been described (Risco *et al*, 2012; Zur *et al*, 2015). Double knockout mice have been used instead of single knockout mice because each of these p38MAPKs normally compensates the loss of the other in many biological processes (Escós *et al*, 2016). WT and p38γ/δ$^{flox/flox}$ mice have been used in all the *in vivo* experiments as control, with similar results. WT, LysM-Cre$^{+/-}$ and p38γ/δ$^{flox/flox}$ mice were used as control when cell recruitment was analysed in the intraperitoneal *C. albicans* model. The recruitment of F4/80$^+$ and Ly6G$^+$ cells in this model was similar in all mouse lines (Appendix Fig S8). Mouse genotypes were determined by PCR. All strains were backcrossed onto the C57BL/6 strain for at least nine generations.

### Antibody

The description of all the antibodies and the dilution used in this study is provided in Appendix Table S1.

### *Candida albicans* infection

*Candida albicans* (strain SC5314) was grown on YPD agar plates at 30°C for 48 h. Eight- to 12-week-old female mice were infected intravenously with $1 \times 10^5$ colony-forming units (CFU) of *C. albicans* and monitored daily for weight and survival. Kidney fungal burden was determined at indicated times post-infection by plating

the kidney homogenates in serial dilutions on YPD agar plates. After 48 h, CFU were counted. Kidneys from infected mice were fixed in 4% formalin and embedded in paraffin. Serial sections were examined microscopically after staining with periodic acid Schiff (PAS) and haematoxylin–eosin.

## Bone marrow-derived macrophages (BMDM) and stimulation

BMDM were isolated and cultured as described in Appendix Supplementary Methods (Risco *et al*, 2012). BMDM were stimulated with: $1 \times 10^6$/ml HK-Ca, 10 μg/ml Zymosan, 5 μg/ml Imiquimod-R837, 200 ng/ml Pam3Cys, 200 ng/ml Poly I-C, 250 ng/ml ODN-1668 (InvivoGen); 10 μg/ml Curdlan (a water-insoluble β-1,3 polysaccharide from *Alcaligenes faecalis*), 100 ng/ml LPS, 100 ng/ml PMA (Sigma). Where indicated, cells were pre-treated with DMSO, SB203580, BIRB0796, C34; PD184352; PRT062607; BI605906; Sorafenib (Bay 43-9006) or NG25. Cells were lysed as described in Appendix Supplementary Methods. For mRNA expression analysis, cells were lysed with NZYol (NZYtech) and RNA extracted using a standard protocol with chloroform–isopropanol–ethanol.

## Neutrophil isolation

Neutrophils were obtained from adult WT and p38γ/δ$^{-/-}$ mice blood followed by hypotonic red blood cell lysis as described in Appendix Supplementary Methods.

## Isolation and stimulation of human monocytes

This protocol is described in Appendix Supplementary Methods. Informed consent was obtained from all healthy volunteers. The experiments conformed to the principles set out in the WMA Declaration of Helsinki and the Department of Health and Human Services Belmont Report.

## Phagocytosis and killing of *Candida albicans* by macrophages and neutrophils

BMDM and neutrophils were obtained, and phagocytosis and killing were performed using the method described earlier (Vonk *et al*, 2012) at a *Candida*/BMDM or *Candida*/neutrophils ratio of 1:10.

## Measurement of reactive oxygen species production

Production of ROS was measured with an assay using luminol as the probe in real time over 220 min. 75,000 neutrophils or 500,000 BMDM or peritoneal cells were plated in 200 μl culture medium (0.05% FBS in HBSS) on a 96-well sterile luminometer plate (Costar, Corning, NY). Cells were stimulated or not as indicated, and L-012 (Wako Chemicals, Osaka, Japan) was incorporated to the medium (7.75 μg/well final concentration) at the beginning of the stimulation. Chemiluminescence was measured at 1-min intervals and expressed as relative light units (RLU).

## Statistical analysis

*In vitro* experiments have been performed at least twice with three independent replicates per experiment. For the analysis of mouse

### The paper explained

#### Problem

*Candida* infections cause high mortality in immunocompromised patients. Sepsis caused by *C. albicans* is one of the most frequent in hospital intensive care units in patients with AIDS or auto-immune diseases and in those undergoing anti-cancer chemotherapy or organ transplantation. Recent studies have shown the important roles of p38γ and p38δ (p38γ/p38δ) in regulating cytokine production, T-cell activation or immune cell recruitment in arthritis and colitis, and in tumorigenesis associated with inflammation. While several studies have demonstrated that p38γ/p38δ are involved in inflammatory processes, the role of these kinases in *C. albicans* infection is completely unknown. Therefore, we hypothesized that p38γ/p38δ might regulate disseminated candidiasis.

#### Results

We show that p38γ/p38δ control cytokine production in response to *C. albicans* in macrophages (mediated by Dectin-1 and TLR receptors) and describe a novel signalling pathway downstream of Dectin-1, which is regulated by p38γ/p38δ. Furthermore, using a mouse model of systemic candidiasis, we found that the deletion of p38γ/p38δ in myeloid cells protected against *C. albicans* infection. Mechanistically, we found that p38γ/p38δ deletion increased antifungal killing capacity of neutrophils and macrophages mediated by increased NOS expression and ROS production in these phagocytes. In addition, p38γ/p38δ deficiency decreased macrophage and neutrophil recruitment to infected kidneys and reduced the production of cytokines and chemokines. We also demonstrate that chemical inhibition of p38γ/p38δ *in vivo* exerts antifungal therapeutic effects in mice infected with *C. albicans*.

#### Impact

Our findings define a major role for p38γ/p38δ in *C. albicans* infection and underscore their importance in regulating inflammatory processes. Our observations point out p38γ/p38δ as potential targets for the development of novel antifungal drugs for human disease and suggest that therapies aimed to inhibit p38γ/p38δ might significantly reduce candidiasis.

survival, production of inflammatory molecules, cell recruitment and CFU in the kidney, the size of the groups was established according to the Spanish ethical legislation for animal experiments. At least 4–5 mice per group were used. Differences in mouse survival were analysed by two-way ANOVA using GraphPad Prism software. Other data were analysed using Student's *t*-test. In all cases, *P*-values < 0.05 were considered significant. Data are shown as mean ± SEM. The exact *n* and *P*-values are given in Appendix Tables S2 and S3.

**Expanded View** for this article is available online.

## Acknowledgements

We thank P. Cohen for critically reading the manuscript. This work was supported by grants from the MINECO [SAF2013-45331-R and SAF2016-79792-R (AEI/FEDER, UE)] to AC and JJS-E, La Marató TV3 Foundation (20133431) to AC and (SAF2014-52009-R) to SA. ERC Consolidator Grant (#310372) and a Spinoza grant of the Netherlands Organization for Scientific Research to MGN, and Wellcome Trust, the Medical Research Council (MRC; UK), the MRC Centre for Medical Mycology at the University of Aberdeen to GDB. DAB and AE receive MINECO FPI fellowships, AR a MINECO Juan de la Cierva award and JD-A a La Caixa Foundation PhD fellowship.

## Author contributions

DA-B, AE, AR, CdF, PF, DG-R, ED-M, JD-A, NA, MAM-S, JJS-E, RZ and AC performed experiments and analysed data; DA-B, MGN and AC designed experiments. GDB, CA, NS, SA and MGN contributed essential reagents; AC wrote the manuscript.

## Conflict of interest

The authors declare that they have no conflict of interest.

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
