## [Review Process File · EMBO Molecular Medicine]

Myeloid cell deficiency of p38g/p38d protects against candidiasis and regulates antifungal immunity

Dayanira Alsina-Beauchamp, Alejandra Escós, Pilar Fajardo, Diego González-Romero, Ester Díaz-Mora, Ana Risco, Miguel A Martín-Serrano, Carlos del Fresno, Jorge Dominguez-Andrés, Noelia Aparicio, Rafal Zur, Natalia Shpiro, Gordon D Brown, Carlos Ardavín, Mihai G Netea, Susana Alemany, Juan J Sanz-Ezquerro and Ana Cuenda

Review timeline:

Submission date:	14 September 2017
Editorial Decision:	12 October 2017
Revision received:	31 January 2018
Editorial Decision:	26 February 2018
Revision received:	12 March 2018
Accepted:	19 March 2018

Editor: Céline Carret

Transaction Report:

1st Editorial Decision

12 October 2017

Thank you for the submission of your manuscript to EMBO Molecular Medicine. We have now heard back from the two referees whom we asked to evaluate your manuscript.

As you will see from the comments below, the data are interesting in principle. However, while the concerns are rather overlapping, referees have a different take on it: one being more positive than the other. Nevertheless, similar concerns are raised and particularly, a deeper understanding of the reported process is suggested, as well as further experiments to increase the translational relevance *in vivo*, along with the provision of missing controls, better statistical analyses, and better documentation of the p38 inhibitor.

We would welcome the submission of a revised version within three months for further consideration and would like to encourage you to address all the criticisms raised as suggested to improve conclusiveness and clarity. Please note that EMBO Molecular Medicine strongly supports a single round of revision and that, as acceptance or rejection of the manuscript will depend on another round of review, your responses should be as complete as possible.

I look forward to receiving your revised manuscript.

***** Reviewer's comments *****

Referee #1 (Remarks for Author):

The manuscript of Alsina-Beauchamp et al. investigates the role of p38 γ /p38 δ in a mouse model of candidiasis. The authors describe a role for Tpl2 in dectin-1 signaling but present data that deletion of p38 γ /p38 δ diminishes fungal infection in vivo independently of TPL2. They present further data that deletion of p38 γ /p38 δ reduced the bacterial load in the kidney, which correlates with decreased neutrophil recruitment and increased iNOS mRNA levels and ROS production. The data represented are often convincing, but statistical significance does not always correlate with the graphical impression (e.g. Fig S3C: Is labeling „ns" always correct? or Fig. 6B: Why is difference between BIRB and SB „ns"?). Therefore, the authors should always show all single values in addition to mean and SD/SE (as done in 3A,B, 6A, S3B, S4C).

In general, the entire characterization of HKC-induced dectin-dependent cytokine production and ERK activation in p38g/d-deficient BMDMs showing ERK-defects is interesting, but becomes irrelevant because of the fact that TPL2 KO shows no phenotype in the Candida model. The observations that cytokine production and recruitment of neutrophils/monocytes to kidney are affected in p38g/d DKO is important and could stand alone as a short report. The final point using small molecule inhibitors, which makes the study interesting for EMBO MM, is focusing exclusively on the comparison between BIRB796 and SB203580, as a means to show p38g/d effect on Candida infection. The relevance of targeting p38g/d against Candida is shown only by looking at the comparison between BIRB796 and SB203580. Independent of the pan-p38 effectiveness of BIRB versus p38a/b inhibition by SB203580, BIRB seems to be a better inhibitor. To show that the specific effectiveness of BIRB versus SB as a readout for the involvement of p38g/d, the authors should do extensive dose response experiments in vitro to make sure that they are working with BIRB796/SB203580 concentration equally effective in suppressing p38a/b signaling. So far, even the SB 203580 control is missing in Fig. S4B. Of course, survival data using BIRB and SB in the mouse model would be even more convincing and would further strengthen the paper.

Other points:

The authors claim that they are probing pp38a in figure 1B, 1E, S1C and S4A. Are these phospho-p38a specific antibodies or are these speculative statements based on abundance?

Fig. 2A: Why is basal ERK1/2 phosphorylation inhibited by LPS, HK-Ca and Zymosan, but not by curdlan?

Fig. 7: This Figure can be omitted. The role of the indicated BIRB inhibition not clear. Does BIRB0796 have the same effect as the DKO or not?

The scientific language and clarity can be improved to avoid misinterpretation and misleading statements:

- page 8, line 13: ``blockaded of ERK1/2.....TLR signaling``. This sounds unsubstantiated. If this is a hypothesis, it could be better stated.
- page 11, TPL2 inhibitor C34 is used at 5 μ M concentration in previous experiments. In line 5, authors state that ``...IL-10 was impaired at high concentrations of C34.....`` This is misleading as 5 μ M is the highest concentration used.
- page 12, first para last line: ``These results indicate that p38g/d, particularly in myeloid cells, regulate kidney infection and survival to C. albicans infection`` should be changed to ``...increased resistance to C.albicans infection`` or ``enhanced survival in response to C.albicans infection``.
- page 14, para 1, last line and data in Fig. 4C: the recruitment of macrophages to peritoneum is only affected in lysMCre model and not in the classical complete KO. This is unusual. In the case of LysM Cre, the Cre insertion actually inactivates the LysM locus, which necessitates the use of Cre containing mice as controls. This control will be necessary to show that there are no defects due to LysM inactivation. Authors should also clarify whether heterozygous LysMCre mice are used for experiments.

- page 15, line 12: ``Fungal phagocytosis by BMDM was similar in both genotypes (~95%)``. Data should be shown in results.

Referee #2 (Comments on Novelty/Model System for Author):

This is an interesting story of high technical quality and novelty. It also addresses an important medical problem using a relevant animal model and therefore has potential therapeutic implications.

Referee #2 (Remarks for Author):

This paper investigates the role of the kinases p38g and p38d in the response to *C. albicans* infection. Evidence is presented that p38g/p38d regulate the production of some cytokines in macrophages incubated with *C. albicans*. Using different stimuli and chemical inhibitors, it is concluded that both the Dectin-1 receptor and the kinase Tpl2 are implicated. Experiments in mice show that p38g/p38d deficiency protects against *C. albicans* infection-associated toxicity, mostly affecting the kidney. Further experiments show that specific downregulation of p38g/p38d in myeloid cells produces the same effect, which correlates with reduced cytokine and chemokine expression as well as with reduced leucocyte infiltration in kidney of the p38g/p38d KO mice. Moreover, downregulation of p38g/p38d increases ROS production and iNOS expression by macrophages and neutrophils. Consistent with the results using genetically-modified mice, a compound that inhibits p38g/p38d protects against *C. albicans*-induced toxicity in mice.

This is an interesting story, mostly supported by high quality data and with potential therapeutic implications. I therefore enthusiastically recommend publication. Below I indicate a few points that should be addressed to strengthen the conclusions.

1. The authors present good evidence indicating that p38g/p38d regulate the production of some cytokines by macrophages incubated with *C. albicans*, and they conclude that Tpl-2 is implicated (Fig 2G). However, Tpl2 does not seem to mediate the *C. albicans* response in mice. It is important to know whether the production of cytokines is affected in Tpl2 deficient mice infected with *C. albicans*. If it was not affected, then Tpl2 does not mediate cytokine production downstream of p38g/p38d by macrophages in response to *C. albicans* in vivo. But if cytokine levels change in the infected Tpl2 KO mice, as for the infected p38g/p38d KO mice, one has to conclude that these changes are not relevant for *C. albicans* pathology. This should be addressed in the manuscript.
2. Is the production of IL6 and TNF α cytokines and MIP-2 and KC chemokines affected in p38g/p38d deficient macrophages treated with *C. albicans*? Or the reduced expression levels detected for these molecules in infected mice are indirectly regulated by p38g/p38d in myeloid cells acting on other cell types?
3. While the results presented show that p38g/p38d in macrophages regulate the inflammatory response in *C. albicans*-treated mice, it is not clear whether the changes in inflammation contribute to the observed toxicity in infected mice. It would be interesting to know the effect of treatment with an anti-inflammatory compound on the toxicity to *C. albicans* in mice.
4. It is proposed that the ability of p38g/p38d to downregulate ROS production and iNOS expression in macrophages and neutrophils may explain the reduced fungi growth in the kidney of p38g/p38d-deficient mice. It is important to provide evidence that the increased levels of ROS and iNOS observed in p38g/p38d KO macrophages and neutrophils indeed contribute to the antifungal activity of these cells against *C. albicans*. At least in the case of ROS, it should be feasible to test whether incubation with anti-oxidants can impair the ability of p38g/p38d macrophages to kill *C. albicans* in vitro, as well as the effect of anti-oxidants on the survival of p38g/p38d deficient mice treated with *C. albicans*.

Minor points:

- Fig 5A, a longer time course of iNOS expression would be useful to know if iNOS is expressed at higher levels or just faster in p38g/p38d deficient BMDM
- Fig 6, panels D and E should be C and D.

- Figure legends should be carefully revised, as key information is often missing. For example, what tissue was analyzed in Fig 4A and 4D? Where was iNOS expression determined in Fig 5H? How long were the mice treated with inhibitors for the WB analysis in Fig S4A and B?

1st Revision - authors' response

31 January 2018

Referee #1 (Remarks for Author):

The manuscript of Alsina-Beauchamp et al. investigates the role of p38γ/p38δ in a mouse model of candidiasis. The authors describe a role for Tpl2 in dectin-1 signaling but present data that deletion of p38γ/p38δ diminishes fungal infection in vivo independently of TPL2. They present further data that deletion of p38γ/p38δ reduced the bacterial load in the kidney, which correlates with decreased neutrophil recruitment and increased iNOS mRNA levels and ROS production. The data represented are often convincing, but statistical significance does not always correlate with the graphical impression (e.g. Fig S3C: Is labeling „ns" always correct? or Fig. 6B: Why is difference between BIRB and SB „ns"?). Therefore, the authors should always show all single values in addition to mean and SD/SE (as done in 3A,B, 6A, S3B, S4C).

We have modified all the figures containing *in vivo* mouse data as the referee asked. Fig 4A-D, Fig 5H, Fig 6B-D and previous Fig S3C, now Fig S4C, show now all single values in addition to mean and SEM. We thank the referee for this suggestion.

In general, the entire characterization of HKC-induced dectin-dependent cytokine production and ERK activation in p38g/d-deficient BMDMs showing ERK-defects is interesting, but becomes irrelevant because of the fact that TPL2 KO shows no phenotype in the Candida model. The observations that cytokine production and recruitment of neutrophils/monocytes to kidney are affected in p38g/d DKO is important and could stand alone as a short report. The final point using small molecule inhibitors, which makes the study interesting for EMBO MM, is focusing exclusively on the comparison between BIRB796 and SB203580, as a means to show p38g/d effect on Candida infection. The relevance of targeting p38g/d against Candida is shown only by looking at the comparison between BIRB796 and SB203580. Independent of the pan-p38 effectiveness of BIRB versus p38a/b inhibition by SB203580, BIRB seems to be a better inhibitor. To show that the specific effectiveness of BIRB versus SB as a readout for the involvement of p38g/d, the authors should do extensive dose response experiments in vitro to make sure that they are working with BIRB796/SB203580 concentration equally effective in suppressing p38a/b signaling. So far, even the SB 203580 control is missing in Fig. S4B. Of course, survival data using BIRB and SB in the mouse model would be even more convincing and would further strengthen the paper.

As the referee points out, it is documented that BIRB796 inhibits p38α/β activity at lower concentration than SB203580 (1,2). We have performed dose response experiments *in vitro* to make sure that SB203580 and BIRB796 suppress p38α/β signalling, as the referee asked. These results are now shown below. We have confirmed that in the mouse macrophage cell line Raw 264.7 stimulated with LPS or curdlan, both SB203580 and BIRB796 inhibits MAPK-KAPK2 and Hsp27 phosphorylation at 1 μM. In Raw 264.7, LPS is much better p38α pathway activator than curdlan. Unfortunately, Heat killed *C. albicans* did not cause phosphorylation of either kinase MAPK-KAPK2 or the heat shock protein Hsp27 in these cells.

Figure legend. p38 α pathway activation is blocked by BIRB796 and SB203580. Raw 264.7 macrophages were incubated for 1 h with or without 0.1, 1 or 10 μ M BIRB976 or SB203580 and then stimulated for 30 min with 100 ng/ml LPS or for 1 h with 10 μ g/ml Curdlan. Cell lysates were immunoblotted with the indicated antibodies. Representative immunoblots from two independent experiments in duplicate are shown. Hsp27 is a physiological MAPKAPK2 substrate and MAPKAPK2 is a kinase specifically phosphorylated by p38 α *in vivo*.

In *in vivo* experiments we have injected three doses of 10 mg inhibitor/kg mouse, which is 80-120 μ l of 12.63 mM BIRB796 (M.W. 526.66) or 17.6 mM SB203580 (M.W. 377.43) per dose. These doses are much higher than 1 μ M; however, it is difficult to compare the *in vitro* with the *in vivo* results since we do not know how much of the inhibitor reached the target organ or how much of the inhibitor is effective in the mouse. Nonetheless, we have shown that (under the conditions used in this study) both compounds SB203580 and BIRB796 inhibit p38 α phosphorylation (Figure S7B) and the production of TNF α in the kidney 24 h post-infection with *C. albicans* (Figure S7A), demonstrating that SB203580 is as effective as BIRB796 *in vivo*.

We also have performed control experiments for previous Figure S4B, now Fig S7C, and show that in the kidney, SB203580 did not affect the phosphorylation of p38 γ and p38 δ induced by *C. albicans* infection.

We agree with the referee that survival data using BIRB796 and SB203580 in the mouse model would further strengthen the paper, however, those experiments are not possible to perform at the present time since both inhibitors block p38 α activity, which plays fundamental physiological roles that affect mouse survival. We hope the publication of this study would encourage pharmaceutical companies to develop potent and specific p38 γ and p38 δ inhibitors.

1. Kuma *et al.* (2005). BIRB796 inhibits all p38 MAPK isoforms *in vitro* and *in vivo*. *J Biol Chem.* 2005 May 20;280(20):19472-9;
2. Bain *et al.*, (2007) The selectivity of protein kinase inhibitors: a further update *Biochem J.* 2007 Dec 15;408(3):297-315)

Other points:

The authors claim that they are probing pp38 α in figure 1B, 1E, S1C and S4A. Are these phospho-p38 α specific antibodies or are these speculative statements based on abundance?

The phospho-p38 antibody is not specific for phospho-p38 α and recognizes all phosphorylated p38 isoforms. We are certain that in Figures 1B, 1E, S1C and S7B (previous Fig S4A) the band recognised by the phospho-p38 antibody is p38 α since in 10% SDS-PAGE, the different p38 isoforms are separated and can be distinguished based on their electrophoretic mobility. From higher to lower mobility: p38 α > p38 β > p38 δ > p38 γ (3-5). In these conditions, anti-phospho-p38 and anti-total p38 α antibodies recognise the same band.

In addition, p38 α is the most abundant p38MAPK isoform in WT macrophages. Its expression is 10 times higher than p38 β , 22 times higher than p38 δ and 300 times higher than p38 γ (6). In p38 γ and p38 δ deficient cells, there is not p38 γ or p38 δ , and the levels of p38 α are 10 times higher than p38 β (6).

3. Zur *et al.*, (2015) Combined deletion of p38 γ and p38 δ reduces skin inflammation and protects from carcinogenesis. *Oncotarget.* 2015 May 30;6(15):12920-35;
4. Criado *et al.*, (2014) Alternative p38 MAPKs are essential for collagen-induced arthritis. *Arthritis Rheumatol.* 2014 May;66(5):1208-17. doi: 10.1002/art.38327;
5. Risco *et al* (2018) p38 γ and p38 δ are involved in T lymphocyte development *Front. Immunol.* doi: 10.3389/fimmu.2018.00065
6. Risco *et al.*, (2012) p38 γ and p38 δ kinases regulate the Toll-like receptor 4 (TLR4)-induced cytokine production by controlling ERK1/2 protein kinase pathway activation. *Proc Natl Acad Sci U S A.* 2012 Jul 10;109(28):11200-5. doi: 10.1073/pnas.1207290109

Fig. 2A: Why is basal ERK1/2 phosphorylation inhibited by LPS, HK-Ca and Zymosan, but not by curdlan?

The answer to the referee's question is that we do not know for sure why this happens, although this observation is frequent in this type of experiments. We could hypothesise that in basal conditions ERK1/2 activation is mediated by "basal pathways" (for example the Ras-Raf pathway, or others) different than the Myd88-TAK1-TPL2 pathway. In response to strong stimuli such as LPS, HK-Ca or Zymosan, the activation of ERK1/2 will shift from the "basal pathways" to mainly the Myd88-TAK1-TPL2 pathway; whereas, in response to a weak stimulus like curdlan, the activation of ERK1/2 is equally mediated by TPL2-dependent and -independent ("basal pathways") pathways. This would explain why in TPL2 deficient cells basal ERK1/2 phosphorylation is not observed after LPS, HK-Ca and Zymosan stimulation, but it is still present in response to curdlan.

Fig. 7: This Figure can be omitted. The role of the indicated BIRB inhibition not clear. Does BIRB0796 have the same effect as the DKO or not?

As the referee suggested, we have now omitted Figure 7 in the new version of the manuscript. We have included new data (Fig S7E) showing that BIRB796 has the same effect than the lack of p38 γ and p38 δ . We found that fungal load in the kidney of WT infected mice treated with BIRB796 is similar to that in the kidney of p38 $\gamma/\delta^{-/-}$ mice infected with *C. albicans*. In addition, BIRB796 treatment does not affect the fungal burden in p38 $\gamma/\delta^{-/-}$ kidney.

The scientific language and clarity can be improved to avoid misinterpretation and misleading statements:

- page 8, line 13: "blockaded of ERK1/2.....TLR signaling". This sounds unsubstantiated. If this is a hypothesis, it could be better stated.

We have deleted that sentence.

- page 11, TPL2 inhibitor C34 is used at 5 μ M concentration in previous experiments. In line 5, authors state that "...IL-10 was impaired at high concentrations of C34....." This is misleading as 5 μ M is the highest concentration used.

The referee is right and we have deleted "...at higher concentration of..." in that sentence.

- page 12, first para last line: "These results indicate that p38g/d, particularly in myeloid cells, regulate kidney infection and survival to C. albicans infection" should be changed to "...increased resistance to C.albicans infection" or "enhanced survival in response to C.albicans infection."

As the reviewer suggested, we have changed the sentence to: "These results indicate that p38 γ/δ , particularly in myeloid cells, increased resistance to *C.albicans* infection"

- page 14, para 1, last line and data in Fig. 4C: the recruitment of macrophages to peritoneum is only affected in LysMCre model and not in the classical complete KO. This is unusual. In the case of LysM Cre, the Cre insertion actually inactivates the LysM locus, which necessitates the use of Cre containing mice as controls. This control will be necessary to show that there are no defects due to LysM inactivation. Authors should also clarify whether heterozygous LysMCre mice are used for experiments.

We have now performed this control experiment and the results are shown in Fig S8. In the intraperitoneal *C. albicans* model, the mouse lines: WT, LysM-Cre^{+/+} and p38 $\gamma/\delta^{\text{flox/flox}}$ show imilar cell recruitment (F4/80⁺ and Ly6G⁺ cells) after infection. We used heterozygous LysMCre mice in this study. This is now stated in the methods section.

page 15, line 12: "Fungal phagocytosis by BMDM was similar in both genotypes (~95%)".

Data should be shown in results.

As the referee asked, we now show this result in Figure S6A.

Referee #2 (Comments on Novelty/Model System for Author):

This is an interesting story of high technical quality and novelty. It also addresses an important medical problem using a relevant animal model and therefore has potential therapeutic implications.

We thank the reviewer for his/her positive assessment.

Referee #2 (Remarks for Author):

This paper investigates the role of the kinases p38g and p38d in the response to *C. albicans* infection. Evidence is presented that p38g/p38d regulate the production of some cytokines in macrophages incubated with *C. albicans*. Using different stimuli and chemical inhibitors, it is concluded that both the Dectin-1 receptor and the kinase Tpl2 are implicated. Experiments in mice show that p38g/p38d deficiency protects against *C. albicans* infection-associated toxicity, mostly affecting the kidney. Further experiments show that specific downregulation of p38g/p38d in myeloid cells produces the same effect, which correlates with reduced cytokine and chemokine expression as well as with reduced leucocyte infiltration in kidney of the p38g/p38d KO mice. Moreover, downregulation of p38g/p38d increases ROS production and iNOS expression by macrophages and neutrophils. Consistent with the results using genetically-modified mice, a compound that inhibits p38g/p38d protects against *C. albicans*-induced toxicity in mice.

This is an interesting story, mostly supported by high quality data and with potential therapeutic implications. I therefore enthusiastically recommend publication. Below I indicate a few points that should be addressed to strengthen the conclusions.

1. The authors present good evidence indicating that p38g/p38d regulate the production of some cytokines by macrophages incubated with C. albicans, and they conclude that Tpl-2 is implicated (Fig 2G). However, Tpl2 does not seem to mediate the C.albicans response in mice. It is important to know whether the production of cytokines is affected in Tpl2 deficient mice infected with C.albicans. If it was not affected, then Tpl2 does not mediate cytokine production downstream of p38g/p38d by macrophages in response to C.albicans in vivo. But if cytokine levels change in the infected Tpl2 KO mice, as for the infected p38g/p38d KO mice, one has to conclude that these changes are not relevant for C. albicans pathology. This should be addressed in the manuscript.

We thank the referee for the suggestion. We have performed those experiments and the results are now shown in Figure S4B. We found that in TPL2^{-/-} mice infected with *C. albicans* the production of cytokines in the kidney was not reduced compared to TPL2^{+/+} mice as we observed in p38γ/δ^{-/-} mice. IL6, TNFα and IL-1β mRNA levels in TPL2^{-/-} kidney were significantly higher than in TPL2^{+/+} mice (Appendix Fig S4B) indicating that *in vivo* TPL2 does not mediate cytokine production downstream of p38γ/p38δ in response to *C.albicans*. These new results are now described and discussed in the new version of the manuscript.

2. Is the production of IL6 and TNFα cytokines and MIP-2 and KC chemokines affected in p38g/p38d deficient macrophages treated with C. albicans? Or the reduced expression levels detected for these molecules in infected mice are indirectly regulated by p38g/p38d in myeloid cells acting on other cell types?

We now show the production of IL-6 and TNFα cytokines and MIP-2 and KC chemokines in p38γ/p38δ deficient macrophages treated with *C. albicans*. These new results are now shown in Fig. 1A. We found that upon HK-Ca stimulation the mRNA expression of TNFα and IL-6 was

not affected by the lack of p38 γ /p38 δ , whereas KC and MIP-2 mRNA expression was markedly reduced in p38 γ / δ -/- BMDM as compared to that in WT BMDM. These new results indicate that p38 γ /p38 δ in myeloid cells acting on other cell types indirectly regulate the reduced expression levels of TNF α and IL-6 in p38 γ / δ -/- infected mice. These new results are now described and discussed in the new version of the manuscript.

3. While the results presented show that p38g/p38d in macrophages regulate the inflammatory response in C. albicans-treated mice, it is not clear whether the changes in inflammation contribute to the observed toxicity in infected mice. It would be interesting to know the effect of treatment with an anti-inflammatory compound on the toxicity to C. albicans in mice.

As the referee suggested, we have performed experiments treating WT and p38 γ / δ -/- infected mice with the antiinflammatory compound ibuprofen. We found that treatment with ibuprofen reduced *C. albicans* load and the recruitment of neutrophils in the kidney of WT mice to similar levels than the loss of p38 γ /p38 δ . In p38 γ / δ -/- mice the treatment with ibuprofen did not affect either fungal burden or the recruitment of neutrophils. In addition, we found that the survival of *C. albicans*-infected WT mice was significantly increased by the treatment with ibuprofen. This protection was more obvious at early times. Ibuprofen, however, did not affect p38 γ / δ -/- mice survival. These new results support that p38 γ /p38 δ are involved in modulating the early beneficial inflammatory response to *C. albicans*. These new results are now shown in Fig S5, and also described and discussed in the new version of the manuscript.

4. It is proposed that the ability of p38g/p38d to downregulate ROS production and iNOS expression in macrophages and neutrophils may explain the reduced fungi growth in the kidney of p38g/p38d-deficient mice. It is important to provide evidence that the increased levels of ROS and iNOS observed in p38g/p38d KO macrophages and neutrophils indeed contribute to the antifungal activity of these cells against C. albicans. At least in the case of ROS, it should be feasible to test whether incubation with anti-oxidants can impair the ability of p38g/p38d macrophages to kill C. albicans in vitro, as well as the effect of antioxidants on the survival of p38g/p38d deficient mice treated with C. albicans.

We thank the referee for the suggestion, and we now provide evidence that the increased levels of ROS observed in p38 γ / δ -/- BMDM contribute to the antifungal activity of these cells against *C. albicans*. We have treated p38 γ / δ -/- BMDM with the anti-oxidant compound Nacetylcysteine (NAC) and found that NAC decreased both ROS levels and *C. albicans* killing *in vitro* in these cells. Also, we have analysed the effect of NAC *in vivo* in the survival of p38 γ / δ -/- mice compared to WT mice. NAC treatment decreased the p38 γ / δ -/- mice survival to similar levels than those in WT mice. These new results are now shown in Fig 5I and Fig S6B-C, and also described and discussed in the new version of the manuscript.

Minor points:

- Fig 5A, a longer time course of iNOS expression would be useful to know if iNOS is expressed at higher levels or just faster in p38g/p38d deficient BMDM

A longer time course of iNOS expression has now been included in Figure 5A. These new data confirm that iNOS expression is higher in p38 γ /p38 δ deficient BMDM than in WT macrophages.

- Fig 6, panels D and E should be C and D.

The labels of the panels in Figure 6 has been corrected

- Figure legends should be carefully revised, as key information is often missing. For example, what tissue was analyzed in Fig 4A and 4D? Where was iNOS expression determined in Fig 5H? How long were the mice treated with inhibitors for the WB analysis in Fig S4A and B?

We have carefully revised figure legends and all the missing information has been now

included.

2nd Editorial Decision

26 February 2018

Thank you for the submission of your revised manuscript to EMBO Molecular Medicine. We have now received the enclosed reports from the referees that were asked to re-assess it. As you will see the reviewers are now globally supportive and I am pleased to inform you that we will be able to accept your manuscript pending minor editorial adjustments including the text minor change mentioned by referee 2.

***** Reviewer's comments *****

Referee #1 (Remarks for Author):

The authors have answered all my questions and provided additional data which further have strengthened the manuscript.

Referee #2 (Remarks for Author):

The revised version includes additional data that address my concerns. Overall the manuscript has been substantially improved and I recommend publication.

A minor point. In page 15, a sentence has been added after describing the new data in Fig S5 that states: "These results support that p38g/p38d are involved in modulating an early beneficial inflammatory response to *C. albicans*." This is probably a mistake and the authors mean deleterious inflammatory response, which would be consistent with the increased survival observed both in p38g/p38d DKO, which have less inflammation, and in iboprufen-treated WT mice.

2nd Revision - authors' response

12 March 2018

Authors made minor editorial changes.

Corresponding Author Name: Ana Cuenda
Journal Submitted to: EMBO Molecular Medicine
Manuscript Number: EMM-2017-08485